# Towards Robust Alignment of Language Models: Distributionally Robustifying Direct Preference Optimization

**Junkang Wu**[1][*] **Yuexiang Xie**[2] **Zhengyi Yang**[1] **Jiancan Wu**[1][†]
**Jiawei Chen**[3] **Jinyang Gao**[2] **Bolin Ding**[2] **Xiang Wang**[1] **Xiangnan He**[4][†]
[1]University of Science and Technology of China   [2]Alibaba Group [3]Zhejiang University
[4]MoE Key Lab of BIPC, University of Science and Technology of China
{jkwu0909, wujcan, xiangnanhe}@gmail.com

## Abstract

This study addresses the challenge of noise in training datasets for Direct Preference Optimization (DPO), a method for aligning Large Language Models (LLMs) with human preferences. We categorize noise into pointwise noise, which includes low-quality data points, and pairwise noise, which encompasses erroneous data pair associations that affect preference rankings. Utilizing Distributionally Robust Optimization (DRO), we enhance DPO's resilience to these types of noise. Our theoretical insights reveal that DPO inherently embeds DRO principles, conferring robustness to pointwise noise, with the regularization coefficient $\beta$ playing a critical role in its noise resistance. Extending this framework, we introduce Distributionally Robustifying DPO (Dr. DPO), which integrates pairwise robustness by optimizing against worst-case pairwise scenarios. The novel hyperparameter $\beta'$ in Dr. DPO allows for fine-tuned control over data pair reliability, providing a strategic balance between exploration and exploitation in noisy training environments. Empirical evaluations demonstrate that Dr. DPO substantially improves the quality of generated text and response accuracy in preference datasets, showcasing enhanced performance in both noisy and noise-free settings. The code is available at https://github.com/junkangwu/Dr_DPO.

## 1 Introduction

Aligning Large Language Models (LLMs) (OpenAI, 2023; Touvron et al., 2023; Anil et al., 2023; Bubeck et al., 2023) with human preferences is critical for their implementation in real-world scenarios. Central to the alignment is the fine-tuning of LLMs using human feedback (Ouyang et al., 2022), ensuring they adhere to human values and mitigate safety risks. Among the alignment methods, Reinforcement Learning from Human Feedback (RLHF) (Ouyang et al., 2022) is becoming a widely adopted technology. It initially learns a reward model on pairwise preference data, and optimizes LLMs using the Proximal Policy Optimization (PPO) (Schulman et al., 2017) method. However, its inherent reinforcement learning nature poses significant challenges to computational efficiency and training stability (Rafailov et al., 2023a; Zhao et al., 2023). Addressing these, Direct Preference Optimization (DPO) (Rafailov et al., 2023a) eschews the explicit reward model learning, using human preferences to train the LLMs directly. It achieves the same objectives (Azar et al., 2023) as RLHF by learning an optimal proxy for each pointwise instance and simultaneously ranking preferences in a pairwise manner, offering greater simplicity and training stability (Ivison et al., 2023).

While offering an effective solution by directly learning a policy from collected data, DPO inevitably heightens the dependency on the data quality (Liu et al., 2023). However, training data is frequently marred by noise, potentially posing a significant challenge to DPO. Here we delineate two primary noise categories based on their origins:

---

[*]Work done at Alibaba Group.
[†]Jiancan Wu and Xiangnan He are the corresponding authors.

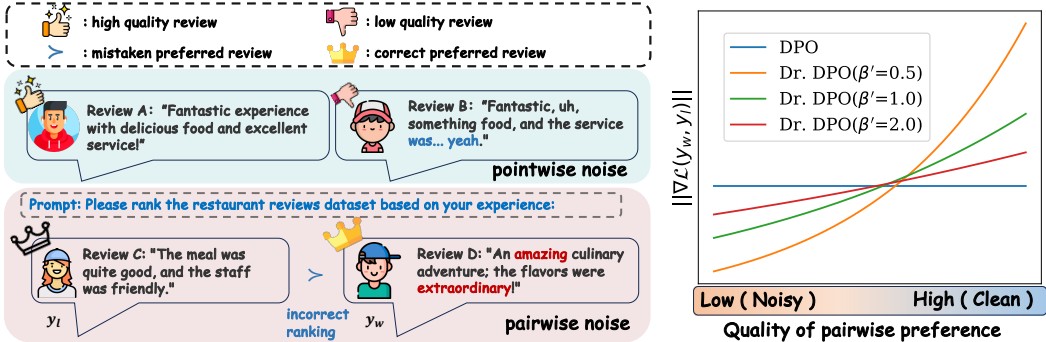

Figure 1: **Left**: An example illustrating pointwise and pairwise noise. **Right**: Comparison of gradients between DPO and Dr. DPO under varying levels of pairwise noise.

- *Pointwise noise* (Gunasekar et al., 2023) refers to low-quality data points containing irrelevant or incoherent information. Taking the movie reviews in Figure 1 (Left) as an example, it might manifest as reviews filled with meaningless chatter, thus rendering them uninformative.

- *Pairwise noise* (Sharma et al., 2023; Cui et al., 2023), on the other hand, arises from erroneous associations between data pairs, leading to misjudged preference rankings. Revisiting the movie reviews in Figure 1 (Left), it is evident in misranked reviews where an inferior review ($y_l$) is incorrectly rated higher than a superior one ($y_w$).

The presence of noisy preferences naturally raises a critical question: *How robust is DPO against pointwise and pairwise noise?* To answer this, we examine DPO through the lens of Distributionally Robust Optimization (DRO) (Namkoong & Duchi, 2017; Duchi & Namkoong, 2018). At the core of DRO is training a model across a distributional family, which is determined by an empirical distribution within a robust radius $\eta$. As a result, DRO endows the model with enhanced robustness *w.r.t.* distributional uncertainty, usually caused by the data noise. By incorporating DRO principles, we can assess the resilience of DPO to the pointwise and pairwise noise. Specifically, our DRO lens on DPO offers insightful findings as follows:

- **DPO is equivalent to applying DRO on the reward function.** The principal contribution of DPO is deriving the optimal policy for PPO in a closed-form expression. This achievement facilitates the implicit determination of a worst-case distribution for optimization, guided by the Kullback-Leibler (KL) divergence criterion. Such an approach endows DPO with intrinsic pointwise robustness, enabling it to explore a better policy model rather than relying solely on the reference model.

- **The DPO's $\beta$ and DRO's $\eta$ share an inverse relationship, highlighting noise levels in the reference model.** Through DRO theory, we establish that higher noise in the reference model necessitates a larger search radius, corresponding to a larger $\eta$ (or equivalently, a smaller $\beta$). This inverse relationship provides a clear measure of the noise level in the reference model.

These findings elucidate the strengths of DPO in ensuring pointwise robustness. Recent effort (Chowdhury et al., 2024) has started addressing pairwise noise in DPO frameworks; however, this method relies on explicit noise estimation, a process that is computationally intensive and may not fully capture noise complexities. Building on these insights, we introduce the *Distributionally Robustifying DPO* (Dr. DPO) [1] framework, aiming to incorporate pairwise robustness within the DPO paradigm. The core idea is optimizing against the worst-case pairwise scenarios, enabling the models to implicitly adjust the importance of data pairs in the gradient space and eliminate the explicit noise estimation. Towards the adjustment, Dr. DPO introduces a simple hyperparameter $\beta' \in (0, +\infty)$ to modulate the loss function, balancing between exploration and exploitation of pairwise preferences. $\beta'$ serves as a pivotal "knob", allowing the navigation from a conservative strategy that diminishes the influence of potentially noisy pairs (*e.g.,* $\beta' = 0.5$) to a risk-tolerant stance that leverages such pairs (*e.g.,* $\beta' = 2$). Consequently, Dr. DPO fosters a more resilient optimization process that effectively mitigates the influence of both pointwise and pairwise noise.

---

[1]The abbreviation "Dr. DPO" not only encapsulates "Distributionally Robustifying DPO" but is playfully intended to echo the abbreviation for "Doctor," adding a quirky element to the naming.

In a nutshell, our contribution is the development of Dr. DPO, which robustifies DPO with just a single additional line of code. Empirical evaluations reveal that Dr. DPO significantly enhances performance across diverse settings, such as controlling the sentiment in generated text and improving the response quality in single-turn dialogues, under both noisy and noise-free conditions.

## 2 PRELIMINARIES

**Bradley-Terry Model.** Given a context $x$ within a finite space of contexts $\mathcal{X}$, we employ the policy $\pi(y|x)$ to independently generate a pair of actions $(y_1, y_2)$. These actions are presented to human raters, who then indicate their preference, with the preferred action labeled as $y_w$ and the less preferred as $y_l$, satisfying $y_w \succeq y_l$. Although we cannot directly observe the latent reward model $r^*(x, y)$ that underlies these preferences, the Bradley-Terry (BT) model (Bradley & Terry, 1952) offers a well-established approach for modeling pairwise comparisons, which is given as:

$$p^*(y_1 \succeq y_2|x) = \frac{\exp(r^*(x, y_1))}{\exp(r^*(x, y_1) + \exp(r^*(x, y_2)))}. \tag{1}$$

Given the dataset $\mathcal{O} = (x^{(i)}, y_w^{(i)}, y_l^{(i)})_{i=1}^N$ sampled from $p^*$, we can parametrize a reward model $r_\phi(x, y)$ and estimate the parameters by optimizing the following logistic regression loss:

$$\mathcal{L}_R(r_\phi, \mathcal{O}) = -\mathbb{E}_{(x, y_w, y_l) \sim \mathcal{O}}[\log \sigma(r_\phi(x, y_w) - r_\phi(x, y_l))], \tag{2}$$

where $\sigma(\cdot)$ is the sigmoid function. As the size of dataset $\mathcal{O}$ grows, the empirical distribution of the dataset $\mathcal{O}$ converges to the underlying distribution $p^*$, and the reward model $r_\phi$ converges to the true reward model $r^*$.

**Reinforcement Learning from Human Feedback (RLHF)** (Ouyang et al., 2022). The standard RLHF paradigm is composed of three phases: i) supervised fine-tuning, ii) reward modeling, and iii) RL fine-tuning. Using the reward model $r_\phi$ learned from the reward modeling, we can then fine-tune the policy $\pi_\theta$ by optimizing the following objective:

$$\max_{\pi_\theta} \mathbb{E}_{x \sim \mathcal{O}, y \sim \pi_\theta(y|x)}[r_\phi(x, y)] - \beta \mathbb{D}_{\text{KL}}[\pi_\theta(y|x)||\pi_{\text{ref}}(y|x)]. \tag{3}$$

In practice, both the language model policy $\pi_\theta$ and the reference policy $\pi_{\text{ref}}$ are typically initialized to the same supervised fine-tuning (SFT) model $\pi_{\text{SFT}}$. Here, $\beta$ is a parameter that controls the strength of the regularization term, and $\mathbb{D}_{\text{KL}}$ represents the KL divergence penalty used to regularize the policy $\pi_\theta$ to be close to $\pi_{\text{ref}}$.

**Directed Preference Optimization (DPO)** (Rafailov et al., 2023a). DPO offers an alternative approach to the RL paradigm described above. It establishes a functional mapping between the reward model and the optimal policy under a KL divergence constraint with the following formulation:

$$r(x, y) = \beta \log \frac{\pi_\theta(y|x)}{\pi_{\text{ref}(y|x)}} + \beta \log Z(x), \tag{4}$$

where $Z(x) = \sum_y \pi_{\text{ref}}(y|x) \exp(r(x, y)/\beta)$ is the partition function. By incorporating this reward into the BT model, the DPO objective enables the comparison of response pairs, facilitating the discrimination between preferred and dispreferred actions, given by:

$$\mathcal{L}_{\text{DPO}}(\pi_\theta; \pi_{\text{ref}}) = -\mathbb{E}_{(x, y_w, y_l) \sim \mathcal{O}}[\log \sigma(\beta \log \frac{\pi_\theta(y_w \mid x)}{\pi_{\text{ref}}(y_w \mid x)} - \beta \log \frac{\pi_\theta(y_l \mid x)}{\pi_{\text{ref}}(y_l \mid x)})]. \tag{5}$$

**Distributionally Robust Optimization (DRO)** (Namkoong & Duchi, 2017; Duchi & Namkoong, 2018). DRO provides a strategic framework to effectively mitigate the uncertainty inherent in training data. It achieves this by optimizing for the worst-case expected loss across a set of potential distributions $Q$. These distributions are confined within a robustness radius $\eta$ anchored around the empirical training distribution $Q_0$, and are bounded by a prescribed divergence metric $\mathbb{D}_\phi$. The formal formulation of DRO can be succinctly expressed as follows:

$$\mathcal{L}_{\text{DRO}} = \max_Q \mathbb{E}_Q[\mathcal{L}(x; \theta)], \qquad s.t. \ \mathbb{D}_\phi(Q, Q_0) \leq \eta, \tag{6}$$

where $\mathcal{L}(x; \theta)$ represents the training loss for an input $x$. Intuitively, models employing DRO exhibit increased robustness due to the presence of $Q$ that acts as an "adversary", optimizing the model under a distribution set with adversarial perturbations instead of a single training distribution.

## 3 ANALYZING DPO'S POINTWISE ROBUSTNESS

In this section, we explore DPO's robustness to pointwise noise, analyzing its response to noise to identify key strengths and vulnerabilities. We assess how noise degrades performance and leverage insights from DRO to understand DPO's underlying resilience mechanisms.

### 3.1 POINTWISE NOISE IMPAIRS DPO PERFORMANCE

We begin by investigating the impact of pointwise noise on DPO through experiments on the IMDB sentiment dataset (Maas et al., 2011). Following the setup in (Havrilla et al., 2023), we fine-tune the GPT-2-large (Radford et al., 2019) model and use SiEBERT (Hartmann et al., 2023), a specialized variant of RoBERTa-large (Liu et al., 2019), for reward calculation. Pointwise noise is introduced exclusively during the SFT stage by incorporating responses generated by the unrefined GPT-2-large model, resulting in lower quality data for this stage, while the data used in the DPO stage remains unchanged. To assess DPO's robustness to this pointwise noise, we evaluate each algorithm by examining the trade-off between the achieved reward and the KL divergence from the reference policy.

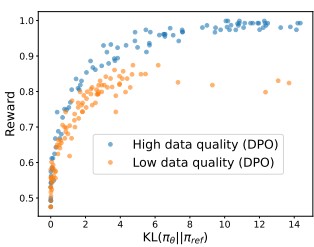

Figure 2: Impact of pointwise noise on the expected reward frontier and KL divergence in DPO ($\beta = 0.1$).

Figure 2 reveals that beyond a KL($\pi_\theta || \pi_{\text{ref}}$) threshold of 10.0, both models converge in terms of reward. Notably, the DPO model trained with high-quality data (blue points) significantly outperforms its low-quality data counterpart (orange points), highlighting the critical impact of data quality on optimizing model performance.

### 3.2 POINTWISE ROBUSTNESS IN REWARD MODELING

In Section 3.1, we explore how pointwise noise negatively affects individual instance rewards. To address this issue and enhance the robustness of LLMs, we propose integrating DRO during the reward modeling stage. We define the Reward Modeling DRO (RM-DRO) objective, which optimizes the expected reward under the worst-case noise distribution within a specified ambiguity set:

$$\max_{\pi_\theta} \mathbb{E}_{x \sim \mathcal{O}, y \sim \pi_\theta(y|x)}[r_\phi(x, y)] \quad s.t. \ \mathbb{D}_\phi(\pi_\theta(y|x), \pi_{\text{ref}}(y|x)) \leq \eta. \tag{7}$$

The direct consequence of pointwise noise is the resultant unreliability of the reference model (SFT). By adopting RM-DRO, we aim to maximize a surrogate objective that accounts for various potential distributions within a robustness radius $\eta$ around the reference distribution $\pi_{\text{ref}}(y|x)$, measured by the distance metric $\mathbb{D}_\phi$. With this formulation, we provide a fresh perspective on DPO.

**A. DPO is Implicitly a Pointwise DRO.**

**Theorem 3.1** (Optimal Reward Function under KL Divergence). *Let the Kullback-Leibler (KL) divergence between policy $\pi_\theta$ and reference policy $\pi_{ref}$ be defined as: $\mathbb{D}_{KL}(\pi_\theta | \pi_{ref}) = \int \pi_\theta(x) \log \left( \frac{\pi_\theta(x)}{\pi_{ref}(x)} \right) dx$. Optimizing the RM-DRO objective as defined in Equation (7) yields an optimal reward $r_{KL}(x, y)$ given by:*

$$r_{KL}(x, y) = \beta^*(\eta) \log \frac{\pi_\theta(y|x)}{\pi_{ref(y|x)}} - \alpha. \tag{8}$$

*Here, $\alpha, \beta$ are Lagrange multipliers, $\beta^*(\eta)$ denotes the optimal value of $\beta$ that minimizes Equation (7), acting as the regularization coefficient in DPO. By deriving the optimal value of $\alpha$, given by:*

$$\alpha^* = -\beta \log \mathbb{E}_{x \sim \mathcal{O}, y \sim \pi_{ref}}[\exp(\frac{r_\theta(y|x)}{\beta})], \tag{9}$$

*Equation 8 can be re-expressed to match the ultimate form of the reward function in Equation 4.*

Please refer to Appendix B.1 for detailed proofs and Appendix B.2 for the formal proof. For a broader discussion on optimal reward functions under general $\phi$-divergences, see Appendix C.1.

Consistent with the reward function formulation in Rafailov et al. (2023a), Theorem 3.1 not only reaffirms established results but also introduces several novel insights, as outlined below:

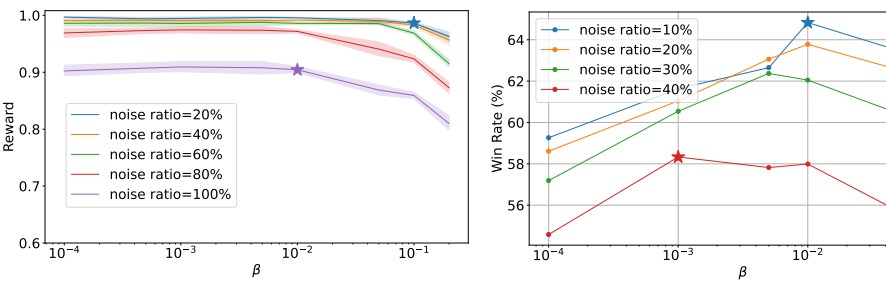

(a) Performance w/ different $\beta$ on IMDB.  (b) Performance w/ different $\beta$ on HH.

Figure 3: (a) Comparative analysis of the effect of pointwise noise on the expected reward frontier for different $\beta$ values on IMDB dataset. (b) Comparative analysis of the effect of pointwise noise on on the win rate for different $\beta$ values on HH dataset. The star ($\star, \star, \star$) indicates the optimal $\beta$ selection for the corresponding pointwise noise ratio.

**Why DPO is Robust to Pointwise Noise.** We propose that the reference distribution closely mirrors the empirical training distribution, given the pre-training step (SFT) common to both RLHF and DPO methods. This ensures the reference distribution in the DPO phase accurately reflects the training data noise. In terms of DRO, while the reference model $\pi_{\text{ref}}$ may not be entirely *reliable*, the implicit robust framework of DPO counters data perturbations effectively. Specifically, the "worst-case distribution" is defined as the distribution that maximizes risk within established divergence constraints, analogous to an adversarial noise model in DRO. Varying $\beta$ enables DPO to exhibit varying search space for a better $\pi_\theta$, leading to improved performance. For more discussion about the connection between DPO and DRO, please refer to Appendix C.2.

Moreover, the incorporation of DRO provides a new interpretation of the coefficient $\beta$ in DPO, transforming it from a mere heuristic design into a "noise reflector". We provide Lemma 3.2 to disclose the relationship between $\beta$ and $\eta$.

**B. The Optimal Value of $\beta$ Reflects the Noise within the SFT Model.**

**Lemma 3.2.** *(Faury et al., 2020, Lemma 5) The optimal $\beta^*(\eta)$ in DPO is monotonically decreasing with respect to $\eta$ and obeys the following relationship:*

$$\beta^*(\eta) = \sqrt{\mathbb{V}_{\pi_{ref}}[r(x,y)]/2\eta}, \tag{10}$$

*where $\mathbb{V}_{\pi_{ref}}[r(x,y)] = \sum_y \pi_{ref}(x,y)(r(y|x) - \sum_y \pi_{ref}(y|x)r(x,y))^2$ denotes the variance of the reward model $r(x,y)$ under the reference distribution $\pi_{ref}$.*

Lemma 3.2 elucidates the inverse correlation between the parameter $\beta$ and the robustness radius $\eta$. Specifically, as noise within the model increases, the required search space expands, necessitating a larger $\eta$ and consequently a smaller optimal $\beta$.

To empirically validate this relationship, we conducted experiments on the IMDB dataset, as outlined in Section 3.1. In these experiments, the noise ratio is controlled by the proportion of low-quality pairs $(y_w, y_l)$ introduced into the training data, generated by the unrefined GPT-2 model. Figure 3a shows that models trained with lower $\beta$ values (e.g., 0.01) outperform those with higher $\beta$ values (e.g., 0.1) when trained on 100% low-quality data. This is because a lower $\beta$ allows for a larger search space to counteract significant pointwise noise in the SFT model.

We also conducted experiments on the HH dataset, injecting pointwise noise during the SFT phase by incorporating rejected responses into the training samples. Importantly, during the DPO phase, the positive and negative samples remained consistent, ensuring noise was introduced only during SFT. The noise ratio is determined by the proportion of rejected responses used as training samples during SFT. As shown in Figure 3b, the optimal value of $\beta$ decreases as the noise ratio increases, indicating that higher noise levels in SFT require a smaller $\beta$ for optimal performance.

For detailed experimental settings and procedures for both datasets, please refer to Appendix C.3, where more comprehensive explanations are provided.

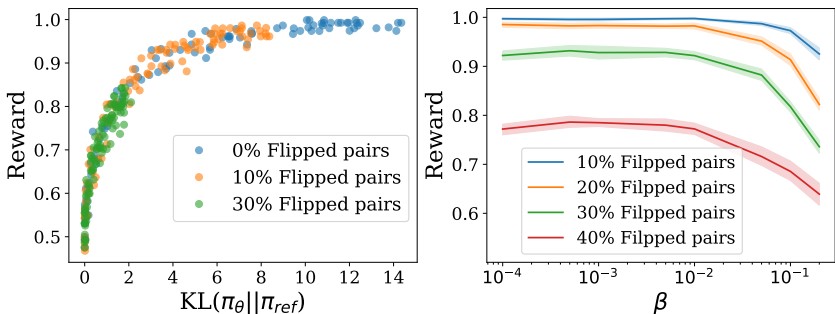

Figure 4: **Left:** Impact of pairwise noise on the expected reward frontier and KL divergence in DPO ($\beta = 0.1$). **Right:** Comparative analysis of the effect of pairwise noise on the expected reward frontier for different $\beta$ values.

## 4 DR. DPO: TOWARD PAIRWISE ROBUSTNESS

In this section, we investigate the impact of pairwise noise and introduce Dr. DPO as a mitigation strategy. We conclude with a theoretical examination of its robustness against such noise.

### 4.1 PAIRWISE NOISE IMPAIRS DPO CONVERGENCE AND PERFORMANCE

We previously explored DPO's pointwise robustness, while recent work (Chowdhury et al., 2024) has examined its resilience to pairwise noise. However, methods that rely on explicit noise estimation may overlook complex noise behaviors. We thus empirically examine how pairwise noise affects DPO's performance. Similar to the experimental settings in Section 3.1, we corrupt the dataset with pairwise noise by randomly swapping pairs $(y_w, y_l)$ in the preference dataset $\mathcal{O} = \{x^{(i)}, y_w^{(i)}, y_l^{(i)}\}_{i=1}^N$. As illustrated in Figure 4 (Left), when exposed to elevated levels of pairwise noise (specifically, 10% and 30% label flipping), DPO exhibits a diminished rate of increase in $\mathrm{KL}(\pi_\theta || \pi_{ref})$ over an equivalent number of training epochs (epoch = 1). This pattern is indicative of a decelerated convergence rate in terms of reward value, especially when compared to the trend of the model trained on noise-free data (0% flipped pairs). Thus the overall performance of DPO drops significantly.

Despite its effectiveness in mitigating pointwise noise through the adjustment of $\beta$, DPO still suffers from the issue of pairwise noise. As shown in Figure 4 (Right), the model trained with 40% flipped data exhibits a significant performance degradation compared to the model trained with noise-free data (0% flipped). Even when the model is trained with a lower $\beta$ value, it still fails to achieve the same level of performance as the model trained with noise-free data. This observation reveals DPO's vulnerability to pairwise noise, motivating enhancements to its robustness against such interference.

### 4.2 DISTRIBUTIONALLY ROBUSTIFYING DPO

Building upon the principles of DRO, we introduce the Distributionally Robustifying DPO (Dr. DPO) framework, designed to enhance DPO's resilience to pairwise noise while preserving its inherent robustness to pointwise noise. The Dr. DPO objective is formulated as follows:

$$\max_{\mathcal{O}'} \mathbb{E}_{(x,y_w,y_l)\sim\mathcal{O}'}[h(x, y_w, y_l)] \quad s.t. \ \mathbb{D}_\phi(\mathcal{O}', \mathcal{O}) \leq \eta'. \tag{11}$$

Here, $h(x, y_w, y_l) = \log \sigma(r_\phi(x, y_w) - r_\phi(x, y_l))$ denotes the log-likelihood objective of dataset point. The $\phi$-divergence, denoted as $\mathbb{D}_\phi(\mathcal{O}', \mathcal{O})$, quantifies the discrepancy between the hypothetical distribution $\mathcal{O}'$ and the dataset distribution $\mathcal{O}$. Additionally, $\eta'$ signifies the robustness radius, which quantifies the degree to which the model can withstand perturbations.

**Theorem 4.1.** *Consider the scenario where the KL divergence is employed to measure the discrepancy between the hypothetical distribution $\mathcal{O}'$ and dataset distribution $\mathcal{O}$, we derive the ultimate loss function for Dr. DPO as follows:*

$$\mathcal{L}_{Dr.\ DPO}(\pi_\theta; \pi_{ref}) = -\beta' \log \mathbb{E}_{\mathcal{O}}[\exp(\frac{h_{DPO}(x, y_w, y_l)}{\beta'})]. \tag{12}$$

*where $h_{DPO}$ represents the log-likelihood in the DPO framework, defined as:*

$$h_{DPO}(x, y_w, y_l) = \log \sigma(\beta \log \frac{\pi_\theta(y_w \mid x)}{\pi_{ref}(y_w \mid x)} - \beta \log \frac{\pi_\theta(y_l \mid x)}{\pi_{ref}(y_l \mid x)}), \tag{13}$$

*with $\beta$ and $\beta'$ being regularization coefficient respectively.*

Please check Appendix B.3 for detailed proofs. In contrast to the objective function described in Equation 7, our Dr. DPO method specifically targets the pairwise noise present within the dataset. Rather than assigning a uniform weight of $\frac{1}{N}$ to each instance $(x^{(i)}, y_w^{(i)}, y_l^{(i)})$ (Rafailov et al., 2023a), Dr. DPO seeks to reweight these instances by an optimal distribution $\mathcal{O}'$. This approach is driven by the intention to capture the incorrect pairs in the data, thereby enhancing the robustness of the resulting policy.

**Theorem 4.2** (Upper Bound for Dr. DPO). *Let $h_{DPO} \in [a, b]$ and $\mathcal{L}_{Dr. DPO}^N$ represents the Dr. DPO loss on $N$ samples. Given a hypothetical distribution $\mathcal{O}'$ satisfying $\mathbb{D}_{KL}(\mathcal{O}', \mathcal{O}) \leq \eta'$ to dataset distribution $\mathcal{O}$, we have that with probability at least $1 - \delta$:*

$$\mathcal{L}_{\mathcal{O}'} \leq \mathcal{L}_{Dr. DPO}^N + \mathcal{B}(\delta, N, \beta'), \tag{14}$$

*where:*

$$\mathcal{B}(\delta, N, \beta') = \frac{2b \exp((b-a)/\beta')}{N - 1 + \exp((b-a)/\beta')} \sqrt{\frac{N}{2} \ln \frac{1}{\delta}}. \tag{15}$$

Please check Appendix B.4 for detailed proofs. In scenarios involving pairwise noise, consider $\mathcal{O}'$ as the "ideal" distribution that discerns the correct ranking between pairwise instances accurately. Theorem 4.2 suggests the ideal loss relative to $\mathcal{O}'$ is upper bounded by the proposed Dr. DPO loss. This bound is achieved when $\mathcal{B}(\delta, N, \beta')$ approaches zero, or in other words, the number of samples $N$ approaches infinity. Furthermore, this upper bound can offer guidance in real-world applications. For instance, in conjunction with Lemma 3.2, we infer that increasing the robustness radius results in a decrease in $\beta'$, consequently increasing $\mathcal{B}(\delta, N, \beta')$. In such a case, to ensure a tight upper bound, it becomes necessary to enlarge the sample size $N$. This relationship between robustness radius, $\beta'$, and $N$ provides insights for guiding the training of LLM models to achieve desired performance.

### 4.3 WHY IS DR. DPO ROBUST TO PAIRWISE NOISE?

Our approach extends the analysis presented in Rafailov et al. (2023a). To understand the resilience of Dr. DPO to pairwise noise, we examine the gradient of its loss function, denoted $\nabla_\theta \mathcal{L}_{Dr. DPO}$:

$$\nabla_\theta \mathcal{L}_{Dr. DPO}(\pi_\theta; \pi_{ref}) = -\beta \mathbb{E}_{(x,y_w,y_l) \sim \mathcal{O}} \big[ \boxed{w(x, y_w, y_l)} \; \boxed{\sigma(\hat{r}_\theta(x, y_l) - \hat{r}_\theta(x, y_w))} \; (\nabla_{\theta,y_w} - \nabla_{\theta,y_l}) \big],$$

$\quad\quad\quad\quad\quad\quad\quad\quad\quad\quad\quad\quad\quad\quad\quad\quad\quad\quad\quad\quad\quad$ Boost mismatched pair gradients.

$\quad\quad\quad\quad\quad\quad\quad\quad\quad\quad\quad\quad$ Reduce incorrect pair's impact.

where $\hat{r}_\theta(x, y) = \beta \log \frac{\pi\theta(y|x)}{\pi_{ref}(y|x)}$ represents the reward function implicitly learned by the policy $\pi_\theta$, relative to a reference policy $\pi_{ref}$. In this framework, $\nabla_{\theta,y_w}$ and $\nabla_{\theta,y_l}$ are gradients increasing the probability of the "chosen" action $y_w$ and decreasing it for the "rejected" action $y_l$, respectively. The factor $\sigma(\hat{r}_\theta(x, y_l) - \hat{r}_\theta(x, y_w))$ serves to amplify the gradient contributions from mismatched action pairs, which is a principal aspect of the Dr. DPO's design aimed at enhancing learning from comparative feedback. Conversely, the function $w(x, y_w, y_l)$, defined as $w(x, y_w, y_l) = \frac{\exp(h(x,y_w,y_l)/\beta')}{\mathbb{E}_{\mathcal{O}}[\exp(h(x,y_w,y_l)/\beta')]}$ (*cf.* Appendix B.5), acts to mitigate the influence of these incorrect pairings. It achieves this by preferentially weighting correct action pairs over incorrect ones, thus refining the policy update mechanism. Moreover, the parameter $\beta'$ does not require intensive tuning; setting it to a default value of 1 typically yields stable enhancements. Remarkably, Dr. DPO is straightforward to implement, requiring only an additional line of code with negligible computational overhead.

**A Toy Example of How Dr. DPO Works.** Consider a training set that contains two samples from both the unperturbed and perturbed datasets. Their corresponding values are assumed to be $[h(x_1, y_{1,w}, y_{1,l}), h(x_2, y_{2,w}, y_{2,l})] = [-0.1, -1.0]$. According to the formula $h(x, y_w, y_l) = \log \sigma(r^+ - r^-)$, the label flip in the second sample leads to $h(x_2, y_{2,w}, y_{2,l}) < \log \sigma(0) = -0.6931$. (1) In the case of Empirical Risk Minimization (ERM), the sum is: $-0.1 + (-1.0) = -1.1$. (2) Under Distributionally Robust Optimization (DRO), when $\beta' = 0.1$, we have the weights

Table 1: Preference accuracy and win-rate comparison on the Anthropic HH dataset with various levels of noise. The columns indicate the performance on both the noise-free dataset and the datasets with inverted response labels, denoted by their respective flip ratios. The performance improvements of cDPO, IPO, and Dr. DPO over DPO are also presented.

| Models | Preference Accuracy | | | | | Win Rate | |
|---|---|---|---|---|---|---|---|
| | 0% Flipped | 10% Flipped | 20% Flipped | 30% Flipped | 40% Flipped | 0% Flipped | 40% Flipped |
| DPO | 63.63 | 62.27 | 61.28 | 58.54 | 55.23 | 54.36 | 49.00 |
| cDPO | $63.48^{-0.23\%}$ | $62.67^{+0.64\%}$ | $61.48^{+0.32\%}$ | $58.69^{+0.27\%}$ | $55.31^{+0.15\%}$ | $51.46^{-5.33\%}$ | $45.68^{-6.78\%}$ |
| IPO | $65.95^{+3.64\%}$ | $64.82^{+4.10\%}$ | $63.52^{+3.65\%}$ | $61.45^{+4.98\%}$ | $56.64^{+2.55\%}$ | $50.81^{-6.53\%}$ | $54.82^{+11.88\%}$ |
| rDPO | $64.37^{+1.16\%}$ | $62.72^{+0.72\%}$ | $62.53^{+2.04\%}$ | $60.56^{+3.45\%}$ | $57.11^{+3.40\%}$ | $52.15^{-4.06\%}$ | $53.54^{+9.26\%}$ |
| Dr. DPO | $\mathbf{66.22}^{+4.07\%}$ | $\mathbf{65.38}^{+5.00\%}$ | $\mathbf{64.19}^{+4.74\%}$ | $\mathbf{62.65}^{+7.02\%}$ | $\mathbf{58.83}^{+6.52\%}$ | $\mathbf{56.67}^{+4.25\%}$ | $\mathbf{61.65}^{+25.81\%}$ |

$[w(x_1, y_{1,w}, y_{1,l}), w(x_2, y_{2,w}, y_{2,l})] \approx [2.0, 0.0]$. Therefore, the outcome is: $(-0.1 \times 2.0) + (-1.0 \times 0.0) = -0.2$, which is greater than $-1.1$. $w(x, y)$ places more emphasis on the unperturbed sample, making the weight allocation more reasonable and also qualifying as a worst-case distribution.

## 5 EXPERIMENTS

In this section, we conduct an empirical assessment of Dr. DPO to evaluate its ability to mitigate noise impacts in preference datasets and to improve performance in noise-free environments. We outline our experimental design, including the datasets used, evaluation metrics, and comparative benchmarks. Our results underscore Dr. DPO's effectiveness, supporting its utility in relevant scenarios.

### 5.1 HOW WELL CAN DR. DPO RESIST THE PAIRWISE NOISE?

**Datasets and Setup.** We conduct experiments on two datasets: IMDB (Maas et al., 2011) and Anthropic HH (Bai et al., 2022). The IMDB dataset is widely utilized for sentiment analysis tasks. The Anthropic HH dataset consists of approximately 170,000 dialogues between humans and automated assistants. The objectives of these experiments were twofold: firstly, to evaluate the robustness of the proposed Dr. DPO against pairwise noise; and secondly, to investigate whether Dr. DPO exhibits superior performance on noise-free datasets. To achieve the first objective, we introduce random inversions between selected and rejected responses in the training data at varying noise levels—specifically, with probabilities of 10%, 20%, 30%, and 40%. For the second objective, we benchmark Dr. DPO against other DPO-related baselines to discern its relative advantages. Unless stated otherwise, all experiments in this study were conducted using the Pythia-2.8B model (Biderman et al., 2023). Please refer to Appendix D for more details.

**Baselines.** We compare Dr. DPO with four baseline methods: (i) The standard DPO, which is the state-of-the-art (SOTA) method for directly optimizing the policy and reward model; (ii) Conservative DPO (cDPO (Rafailov et al., 2023b)), a variant introduced by the authors of DPO to address scenarios with probabilistically flipped labels by incorporating a binary cross-entropy (BCE) loss; and (iii) IPO (Azar et al., 2023), an innovative approach that enables learning directly from preferences, bypassing both the reward modeling phase and reliance on the BT model. (iv) rDPO (Chowdhury et al., 2024) is a variant of DPO that de-biases the effect of preference noise and makes the policy robust.

**Metrics.** We adopt two metrics, *Preference Accuracy*, and *Win-Rate*, in the experiments. The *Preference Accuracy* measures the proportion of test instances from the Anthropic HH dataset where the model's predicted reward for the preferred response, $r(x, y_w)$, exceeds that of the less preferred alternative, $r(x, y_l)$. We further draw on the evaluation framework introduced by Rafailov et al. (2023a) to calculate the *Win Rate*. This metric assesses how often the GPT-4 model selects a response generated by our model rather than the chosen response within the dataset. The Win-Rate computation is specifically designed for the single-turn dialogue portion of HH dataset's test subset.

**Dr. DPO Outperforms Across Noise Levels.** Table 1 illustrates the effect of noise on preference accuracy, showing a decrease from 63.63% to 55.23% as the label flip ratio rises from 0% to 40%. This indicates a deteriorated ability to distinguish between responses with increasing noise. In a noise-free environment, cDPO achieves a preference accuracy of 63.48%, which falls to 55.31% at a 40% noise level, highlighting the limited impact of BCE loss adjustments. In comparison, the IPO method consistently outperforms DPO by an average of 3.78% in accuracy, proving its efficacy.

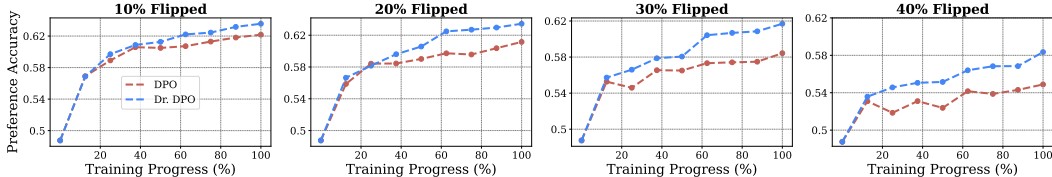

Figure 5: MT-Bench evaluates DPO and its variants using GPT-4, showing Win, Tie, and Loss rates at 0% and 40% pairwise noise levels in Figures 1-2. Figure 3 illustrates Dr. DPO's win rate across different $\phi$-divergences, while Figure 4 presents its Preference accuracy for varying $\beta'$ values.

Figure 6: Preference accuracy across training steps for different levels of pairwise noise.

rDPO reports a 3% accuracy gain over DPO. Notably, Dr. DPO exhibits outstanding noise resistance, achieving the highest preference accuracies of 66.22% in the absence of noise and 58.83% with 40% noise, superior to DPO, cDPO, IPO, and rDPO.

**Dr. DPO Surpasses Other Methods.** We evaluated the response quality of various models—DPO, cDPO, IPO, rDPO, and Dr. DPO—across different noise levels. Table 1 shows that in a noiseless setting, DPO leads with a 56.67% win rate. However, as noise increases, both DPO and cDPO's performance declines, whereas Dr. DPO excels, achieving a 61.65% win rate.

**Dr. DPO's Performance in the 0% Flipped Case.** The 0% flipped case indicates no intentional label flips were introduced, representing the original dataset. Nevertheless, this does not mean the dataset is free of label noise. Dr. DPO continues to outperform DPO and IPO in this scenario, suggesting the presence of inherent label noise in the dataset. This observation aligns with rDPO's findings (Chowdhury et al., 2024), where a default flip rate of 0.1 improved performance on the HH dataset, further justifying the design of Dr. DPO.

## 5.2 COMPARING DR. DPO WITH BASELINES ON MT-BENCH

To evaluate the generation quality of DPO and its variants, we conduct pairwise comparisons using the MT-Bench framework (Zheng et al., 2023). This framework, grounded in GPT-4, reliably aligns with human evaluative preferences, exhibiting an agreement rate exceeding 80% on the quality of outputs from LLMs. Adhering to the established MT-Bench guidelines (Zheng et al., 2023) [2], our approach involves generating model responses at a controlled temperature of 0.7 and restricting the token count to a maximum of 1024. We systematically compare the outputs from the base DPO model and its variants, which have been fine-tuned with 0% and 40% flipped pairs on the HH dataset.

Figure 5 (1,2) shows that Dr. DPO consistently outperforms DPO in both noise-free and noisy datasets, becoming the only method to exceed DPO's performance in the MT-Bench evaluation. While IPO slightly improves over DPO in the noise-free dataset, it underperforms in the noisy dataset where DPO prevails. In contrast, Dr. DPO demonstrates robust, significant enhancements in both conditions, highlighting its superior ability to generate high-quality responses.

## 5.3 ABLATION STUDIES ON DR. DPO

We conduct ablation studies to investigate the impact of the $\phi$-divergence and $\beta'$ on the performance of Dr. DPO, and provide the convergence analysis. More ablation studies are listed in Appendix D.

---
[2] https://github.com/lm-sys/FastChat/blob/main/fastchat/llm_judge/gen_model_answer.py

Table 2: Comparison of KL Divergence and Win Rate of Model Performance at Different Noise Ratios

| Noise Ratio | Loss Function | Win Rate | KL | $\beta$ |
|---|---|---|---|---|
| **0%** | DPO | 54.36 | 19.68 | 0.1 |
| | Dr.DPO | 56.67 | 20.01 | 0.1 |
| **40%** | DPO | 49.00 | 8.32 | 0.1 |
| | Dr.DPO | 61.65 | 7.96 | 0.1 |

Table 3: Comparison of Model Performance via Win Rate on the LLaMA2-13B.

| | Loss Function | Win Rate |
|---|---|---|
| Llama2-13b | DPO | 59.05 |
| | cDPO | 53.52 |
| | rDPO | 60.54 |
| | Dr.DPO | 63.67 |

**Evaluating the Impact of $\phi$-divergence on Dr. DPO.** Figure 5 (3) explores Dr. DPO's performance with various $\phi$-divergences, including Jensen-Shannon (JS) and $\alpha$-divergence. Demonstrated results indicate that Dr. DPO consistently outperforms the baseline DPO when $\beta'$ is set to 1.0, serving as a viable default without requiring further adjustments. This is in contrast to the baseline $\beta = 0.1$ setting, where although improvements can be realized by manually tuning $\beta^*$, the process becomes time-consuming and impractical for regular use.

**Evaluating the Impact of $\beta'$.** Figure 5 (4) illustrates how varying $\beta'$ across different noise levels affects preference accuracy. The experimental results reveal a trend wherein increased noise levels correspond to a reduced optimal value for $\beta'$, which is consistent with our theoretical analysis provided in Section 3. Consequently, we propose a default setting of $\beta' = 1.0$ for balancing accuracy and robustness in the presence of noise.

**Convergence Analysis.** Figure 6 shows that Dr. DPO not only converges faster but also surpasses DPO in the early training stage, attributed to its superior management of flipped noisy pairs. This enhancement meets our goal of boosting DPO's robustness in noisy environments.

**Dr. DPO does not increase KL divergence.** A potential concern is that Dr. DPO might improve performance by increasing the KL divergence. To address this, we computed the KL divergence of models trained under two noise ratios: 0% and 40%. As shown in Table 2, the KL divergence does not increase with DRDPO for the same $\beta$ settings. This comparison demonstrates that DRDPO maintains a stable KL divergence, effectively mitigating the concern that performance gains are achieved through increased divergence.

**Comparing Dr. DPO with Baselines on LLaMA2-13B.** We further evaluate the performance of Dr. DPO on the LLaMA2-13B dataset, which is a large-scale language model. Table 3 shows that Dr. DPO outperforms DPO, cDPO, and rDPO, achieving a win rate of 63.67% compared to 59.05% for DPO, 53.52% for cDPO, and 60.54% for rDPO. These results demonstrate the superior performance of Dr. DPO in generating high-quality responses on the LLaMA2-13B dataset.

## 6   DISCUSSION

**Conclusion.** In this study, we analyze DPO's robustness from a DRO perspective, highlighting its resilience to pointwise noise. We establish a link between DPO's regularization and DRO's robustness, showing that a smaller regularization parameter $\beta$ enhances stability against uncertain data. Our experiments confirm the crucial role of $\beta$ in noise resistance but uncover DPO's weakness against pairwise noise. To address this, we introduce a novel Distributionally Robustifying DPO framework with an additional parameter $\beta'$ that balances data pair importance in training to enhance model robustness. The Dr. DPO's fine-tuning of exploration and exploitation could markedly improve the alignment of language models, assuring reliable performance in the presence of real-world noise.

**Limitations and future work.** The current work introduces Dr. DPO, an enhancement to DPO that addresses label flipping noise in training datasets through an additional hyperparameter $\beta'$. Despite the robust performance indicated by empirical results with a default $\beta'$ value of 1.0, the need for parameter tuning in different applications remains. The sensitivity of $\beta'$ to data and task specifics may require a search process to fully leverage Dr. DPO's potential. Additionally, we aim to explore adaptive mechanisms that allow Dr. DPO to adjust robustness dynamically during training, reducing the need for manual parameter tuning and enhancing its practical applicability across diverse tasks.

ACKNOWLEDGMENTS

This research is supported by the National Science and Technology Major Project (2023ZD0121102), National Natural Science Foundation of China (92270114, U24B20180, 62121002, 62302321). This research was also supported by the advanced computing resources provided by the Supercomputing Center of the USTC.

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

## A  RELATED WORK

**Reinforcement Learning from Human Feedback.** RLHF (Christiano et al., 2017; Bai et al., 2022; Touvron et al., 2023; Ouyang et al., 2022) has emerged as a key method for aligning language models with human values and preferences, mitigating the generation of biased or factually incorrect outputs. Compared to supervised learning, RLHF is rather complex, less stable, and requires more memory resources. These challenges have motivated the development of alternatives to the RLHF pipeline. For example, RAFT (Dong et al., 2023) uses an existing reward model to select the best set of training samples based on the model outputs, while RRHF (Yuan et al., 2023) leverages a much simpler ranking loss to align human preferences and retain the performance of PPO. DPO (Rafailov et al., 2023a) is another alternative to RLHF that uses a preference loss function to directly optimize the LLMs, and has been shown to be more stable and less computationally intensive than RLHF. Despite these efforts, all the methods ignore the noise in the training data, which can lead to suboptimal performance. In a recent advancement, WPO has been introduced as a novel strategy to simulate on-policy learning with off-policy preference data, effectively addressing the distributional gap problem and enhancing the optimization process without incurring additional costs (Zhou et al., 2024).

**Distributionally Robust Optimization.** DRO differs from traditional robust optimization methods (Jin et al., 2020; Lian et al., 2020; Wu et al., 2023) by minimizing the worst-case error within an uncertainty set defined by constraints like $\phi$-divergence (Namkoong & Duchi, 2017; Duchi & Namkoong, 2018), Wasserstein distance (Shafieezadeh-Abadeh et al., 2015; Sinha et al., 2018; Huang et al., 2022), and shape (Lam et al., 2021; Chen et al., 2021). (Michel et al., 2021; 2022) introduced parametrization to the uncertainty set for greater architectural flexibility. Separately, (Zhai et al., 2021) addressed sensitivity to outliers in DRO, diverging from the other studies. And (Oren et al., 2019) introduced a novel approach to language modeling that enhances robustness by optimizing against the worst-case topic mixture, aligning with the overarching theme of minimizing maximum losses within predefined uncertainty sets.

## B  APPENDIX OF PROOFS

### B.1  PROOF OF THEOREM 3.1

**Theorem 3.1** (Optimal Reward Function under KL Divergence). *Let the Kullback-Leibler (KL) divergence between policy $\pi_\theta$ and reference policy $\pi_{ref}$ be defined as:* $\mathbb{D}_{KL}(\pi_\theta|\pi_{ref}) = \int \pi_\theta(x) \log\left(\frac{\pi_\theta(x)}{\pi_{ref}(x)}\right) dx$. *Optimizing the RM-DRO objective as defined in Equation (7) yields an optimal reward $r_{KL}(x,y)$ given by:*

$$r_{KL}(x,y) = \beta^*(\eta) \log \frac{\pi_\theta(y|x)}{\pi_{ref(y|x)}} - \alpha. \tag{8}$$

*Here, $\alpha, \beta$ are Lagrange multipliers, $\beta^*(\eta)$ denotes the optimal value of $\beta$ that minimizes Equation (7), acting as the regularization coefficient in DPO. By deriving the optimal value of $\alpha$, given by:*

$$\alpha^* = -\beta \log \mathbb{E}_{x \sim \mathcal{O}, y \sim \pi_{ref}}[\exp(\frac{r_\theta(y|x)}{\beta})], \tag{9}$$

*Equation 8 can be re-expressed to match the ultimate form of the reward function in Equation 4.*

*Proof.*

**Definition B.1** ($\phi$-divergence (Nguyen et al., 2010))**.** For any convex function $\phi$ with $\phi(1) = 0$, the $\phi$-divergence between $Q$ and $Q_0$ is:

$$D_\phi(\pi_\theta, \pi_{\text{ref}}) := \mathbb{E}_{\pi_{\text{ref}}}[\phi(\frac{\pi_\theta}{\pi_{\text{ref}}})] \tag{16}$$

where $D_\phi(Q, Q_0) = \infty$ if $Q$ is not absolutely continuous with respect to $Q_0$. Specially, when $\phi(x) = x \log x - x + 1$, $\phi$-divergence degenerates to the well-known KL divergence.

**Definition B.2** (Convex conjugate (Hiriart-Urruty & Lemaréchal, 2004))**.** We consider a pair $(A, B)$ of topological vector spaces and a bilinear form $\langle \cdot, \cdot \rangle \to \mathbb{R}$ such that $(A, B, \langle \cdot, \cdot \rangle)$ form a dual pair. For a convex function $f : \mathbb{R} \to \mathbb{R}$, $dom f := \{x \in \mathbb{R} : f(x) < \infty\}$ is the effective domain of $f$. The convex conjugate, also known as the Legendre-Fenchel transform, of $f : A \to \mathbb{R}$ is the function $f^* : B \to \mathbb{R}$ defined as

$$f^*(b) = \sup_a \{ab - f(a)\}, \quad b \in B \tag{17}$$

**Theorem B.3** (Interchange of minimization and integration (Ben-Tal & Teboulle, 2007))**.** *Let $(\Omega, \mathcal{F})$ be a measurable space equipped with $\sigma$-algebra $\mathcal{F}$, $L^p(\Omega, \mathcal{F}, P)$ be the linear space of measurable real valued functions $f : \Omega \to \mathbb{R}$ with $||f||_p < \infty$, and let $\mathcal{X} := L^p(\Omega, \mathcal{F}, P)$, $p \in [1, +\infty]$. Let $g : \mathbb{R} \times \Omega \to \mathbb{R}$ be a normal integrand, and define on $\mathcal{X}$. Then,*

$$\min_{x \in \mathcal{X}} \int_\Omega g(x(\omega), \omega) \, dP(\omega) = \int_\Omega \min_{s \in \mathbb{R}} g(s, \omega) \, dP(\omega) \tag{18}$$

To ease the derivation, we denote the likelihood ratio $L(y|x) = \pi_\theta(y|x)/\pi_{\text{ref}}(y|x)$. Note that the $\phi$-divergence between $\pi_\theta$ and $\pi_{\text{ref}}$ is constrained, and thus $L(.)$ is well-defined. For brevity, we usually short $L(y|x)$ as $L$. And in terms of Definition B.1 of $\phi$-divergence, the expression of RM-DRO (*cf.* Equation equation 7) becomes:

$$\mathcal{L}_{\text{RM-DRO}}^\phi = \max_L \mathbb{E}_{x \sim \mathcal{D}, y \sim \pi_{\text{ref}}}[r_\theta(y|x)L] \qquad s.t. \; \mathbb{E}_{\pi_{\text{ref}}}[\phi(L(y|x))] \leq \eta \tag{19}$$

Note that $\mathbb{E}_{\pi_{\text{ref}}}[r_\theta(y|x)L]$ and $\mathbb{E}_{\pi_{\text{ref}}}[\phi(L(y|x))]$ are both convex in $L$. We use the Lagrangian function solver:

$$\mathcal{L}_{\text{RM-DRO}}^\phi = \min_{\beta \geq 0, \alpha} \max_{L(y|x)} \left\{ \mathbb{E}_{x \sim \mathcal{D}, y \sim \pi_{\text{ref}}}[r_\theta(y|x)L(y|x)] - \beta[\mathbb{E}_{\pi_{\text{ref}}}[\phi(L(y|x))] - \eta] + \alpha(\mathbb{E}_{\pi_{\text{ref}}}[L(y|x)] - 1) \right\}$$

$$= \min_{\alpha \geq 0, \beta} \left\{ \beta\eta - \alpha + \beta \max_{L(y|x)} \left\{ \mathbb{E}_{x \sim \mathcal{D}, y \sim \pi_{\text{ref}}}[\frac{r_\theta(y|x) + \alpha}{\beta}L(y|x) - \phi(L(y|x))] \right\} \right\}$$

$$= \min_{\beta \geq 0, \alpha} \left\{ \beta\eta - \alpha + \beta\mathbb{E}_{x \sim \mathcal{D}, y \sim \pi_{\text{ref}}}[\max_{L(y|x)} \left\{ \frac{r_\theta(y|x) + \alpha}{\beta}L(y|x) - \phi(L(y|x)) \right\}] \right\}$$

$$= \min_{\beta \geq 0, \alpha} \left\{ \beta\eta - \alpha + \beta\mathbb{E}_{x \sim \mathcal{D}, y \sim \pi_{\text{ref}}}[\phi^*(\frac{r_\theta(y|x) + \alpha}{\beta})] \right\} \tag{20}$$

The first equality holds due to the strong duality (Boyd & Vandenberghe, 2004). The second equality is a re-arrangement for optimizing $L(y|x)$. The third equation follows by the Theorem B.3. The last equality is established based on the definition of convex conjugate B.2. When we choose KL-divergence, we have $\phi_{\text{KL}}(x) = x \log x - x + 1$. It can be deduced that $\phi_{\text{KL}}^*(x) = e^x - 1$. Then, we have:

$$\mathcal{L}_{\text{RM-DRO}}^{KL} = \min_{\beta \geq 0, \alpha} \left\{ \beta\eta - \alpha + \beta\mathbb{E}_{x \sim \mathcal{D}, y \sim \pi_{\text{ref}}}[\phi^*(\frac{r_\theta(y|x) + \alpha}{\beta})] \right\}$$

$$= \min_{\beta \geq 0, \alpha} \left\{ \beta\eta - \alpha + \beta\mathbb{E}_{x \sim \mathcal{D}, y \sim \pi_{\text{ref}}}[\exp(\frac{r_\theta(y|x) + \alpha}{\beta}) - 1]] \right\} \tag{21}$$

and the maximum of term $L(y|x)$ in Equation equation 20 is achieved when

$$L(y|x) = \exp(\frac{r_\theta(y|x) + \alpha}{\beta}). \tag{22}$$

We differentiate the Lagrangian function *w.r.t.* $\alpha$ and set it to zero:

$$\frac{\partial}{\partial\alpha}\left\{\beta\eta - \alpha + \beta\mathbb{E}_{x\sim\mathcal{D},y\sim\pi_{\text{ref}}}[\exp(\frac{r_\theta(y|x)+\alpha}{\beta}) - 1]\right\} = 0 \tag{23}$$

Then, we have:

$$\alpha^* = -\beta\log\mathbb{E}_{x\sim\mathcal{D},y\sim\pi_{\text{ref}}}[\exp(\frac{r_\theta(y|x)}{\beta})] \tag{24}$$

Substituting the optimal $\alpha^*$ into the Lagrangian function equation 21, we have:

$$\begin{aligned}
\mathcal{L}^{KL}_{\text{RM-DRO}} &= \min_{\beta\geq 0,\alpha}\left\{\beta\eta - \alpha + \beta\mathbb{E}_{x\sim\mathcal{D},y\sim\pi_{\text{ref}}}[\exp(\frac{r_\theta(y|x)+\alpha}{\beta}) - 1]\right\} \\
&= \min_{\beta\geq 0}\left\{\beta\eta + \beta\log\mathbb{E}_{x\sim\mathcal{D},y\sim\pi_{\text{ref}}}[\exp(\frac{r_\theta(y|x)}{\beta})]\right\} \\
&= \beta^*(\eta)\log\mathbb{E}_{x\sim\mathcal{D},y\sim\pi_{\text{ref}}}[\exp(\frac{r_{\text{DPO}}(x,y)}{\beta^*(\eta)})] + C,
\end{aligned} \tag{25}$$

where $\beta^*(\eta)$ signifies the optimal value of $\beta$ that minimizes the Lagrangian function and $C = \beta\eta$. Besides, if we plug the optimal $\alpha^*$ into the optimal $L(y|x)$, we have:

$$\begin{aligned}
L^*(y|x) &= \exp(\frac{r_\theta(y|x)+\alpha^*}{\beta^*}) \\
&= \exp(\frac{r_\theta(y|x)}{\beta^*})\frac{1}{Z(x)}
\end{aligned} \tag{26}$$

where $Z(x) = \mathbb{E}_{x\sim\mathcal{D},y\sim\pi_{\text{ref}}}[\exp(\frac{r_\theta(y|x)}{\beta})]$. Here, we rearrange Equation equation 26 and obtain the expression of $r_{\text{KL}}(x,y)$:

$$r_{\text{KL}}(x,y) = \beta^*\log L^*(y|x) + \beta\log Z(x) = \beta^*\log\frac{\pi_\theta}{\pi_{\text{ref}}} + \beta\log Z(x) \tag{27}$$

The theorem is proven. In comparison to the proofs presented in DPO (Rafailov et al., 2023a), our proof is comprehensive and direct, applicable to any $\phi$-divergence constraints in the general PPO objective. DPO (Rafailov et al., 2023a) represents a specific case that employs strategies to construct an objective in the form of KL-divergence.

$\square$

### B.2 FORMAL PROOF

**Step 1: Definition of $\phi$-divergence** The $\phi$-divergence between $\pi_\theta$ and $\pi_{\text{ref}}$ is defined as:

$$D_\phi(\pi_\theta, \pi_{\text{ref}}) = \mathbb{E}_{\pi_{\text{ref}}}\left[\phi\left(\frac{\pi_\theta(y|x)}{\pi_{\text{ref}}(y|x)}\right)\right],$$

where $\phi$ is a convex function with $\phi(1) = 0$. For KL-divergence, $\phi(t) = t \log t - t + 1$.

**Step 2: Reformulating the RM-DRO Objective** Using the Lagrangian dual form to incorporate the robustness constraint, the RM-DRO objective becomes:

$$\min_{\beta \geq 0, \alpha} \max_{L(y|x)} \left\{ \mathbb{E}_{x \sim \mathcal{D}, y \sim \pi_{\text{ref}}}[r_\theta(y|x)L(y|x)] - \beta[\mathbb{E}_{\pi_{\text{ref}}}[\phi(L(y|x))] - \eta] + \alpha(\mathbb{E}_{\pi_{\text{ref}}}[L(y|x)] - 1) \right\}$$

where $\beta, \alpha$ are the Lagrange multipliers associated with the robustness constraint, $L(y|x) = \pi_\theta(y|x)/\pi_{\text{ref}}(y|x)$.

**Step 3: Solving for Optimal Policy** According to the proof of Theorem 3.1, solving this Lagrange function with KL-divergence yields:

$$r_\phi(x, y) = \beta \log \frac{\pi_\theta(y|x)}{\pi_{\text{ref}}(y|x)} - \alpha,$$

where $\alpha$ is a normalization constant ensuring that $\pi_\theta$ is a valid probability distribution:

$$\alpha = -\beta \log \mathbb{E}_{y \sim \pi_{\text{ref}}}\left[\exp\left(\frac{r_\phi(x, y)}{\beta}\right)\right].$$

**Step 4: Connection to DPO** The reward function derived above matches the closed-form expression for the reward function in DPO:

$$r(x, y) = \beta \log \frac{\pi_\theta(y|x)}{\pi_{\text{ref}}(y|x)} + \beta \log Z(x),$$

where $Z(x)$ is the partition function defined as:

$$Z(x) = \sum_y \pi_{\text{ref}}(y|x) \exp\left(\frac{r(x, y)}{\beta}\right).$$

**Step 5: Robustness to Pointwise Noise** The DRO formulation inherently considers the worst-case distribution within the robustness radius $\eta$. By solving the constrained optimization problem under KL-divergence, DPO implicitly optimizes for the worst-case distributional perturbation. Thus, DPO achieves robustness to pointwise noise, as the DRO mechanism mitigates the impact of noisy data points by optimizing over a family of perturbed distributions.

By aligning the RM-DRO objective with the reward function formulation in DPO, we establish that DPO implicitly operates as a pointwise DRO framework under KL-divergence. This demonstrates its inherent robustness to pointwise noise.

### B.3 PROOF OF THEOREM 4.1

**Theorem 4.1.** *Consider the scenario where the KL divergence is employed to measure the discrepancy between the hypothetical distribution $\mathcal{O}'$ and dataset distribution $\mathcal{O}$ , we derive the ultimate loss function for Dr. DPO as follows:*

$$\mathcal{L}_{Dr.\ DPO}(\pi_\theta; \pi_{ref}) = -\beta' \log \mathbb{E}_{\mathcal{O}}[\exp(\frac{h_{DPO}(x, y_w, y_l)}{\beta'})]. \tag{12}$$

*where $h_{DPO}$ represents the log-likelihood in the DPO framework, defined as:*

$$h_{DPO}(x, y_w, y_l) = \log \sigma(\beta \log \frac{\pi_\theta(y_w \mid x)}{\pi_{ref}(y_w \mid x)} - \beta \log \frac{\pi_\theta(y_l \mid x)}{\pi_{ref}(y_l \mid x)}), \tag{13}$$

*with $\beta$ and $\beta'$ being regularization coefficient respectively.*

*Proof.* To solve the optimization problem in Equation equation 11, we first introduce the Lagrangian function:

$$\begin{aligned} \mathcal{O}(\mathcal{O}', \beta', \alpha') &= \mathbb{E}_{(x,y_w,y_l) \sim \mathcal{O}'}[h(x, y_w, y_l)] \\ &+ \beta'(\mathbb{D}_\phi(\mathcal{O}', \mathcal{O}) - \eta') + \alpha'(\mathbb{E}_{\mathcal{O}}[\frac{\mathcal{O}'}{\mathcal{O}}] - 1). \end{aligned} \tag{28}$$

Then, we can obtain the optimal distribution $\mathcal{O}'^{,*}$ by solving the following saddle-point problem:

$$\mathcal{O}'^{,*} = \arg\max_{\mathcal{O}'} \min_{\beta',\alpha'} \mathcal{O}(Q, \beta', \alpha'). \tag{29}$$

Specifically, when the KL divergence is selected as the measure of $\phi$-divergence, that is, $\mathrm{KL}(\mathcal{O}', \mathcal{O}) = \sum_{i=1}^{N} \mathcal{O}'_i \log(\mathcal{O}'_i/\mathcal{O}_i)$, the optimal distribution $\mathcal{O}'^{,*}_{\mathrm{KL}}$ can be derived as follows:

$$\mathcal{O}'^{,*}_{\mathrm{KL}} = \frac{1}{Z^*} \exp\left(\frac{h(x, y_w, y_l)}{\beta'}\right), \tag{30}$$

where $Z^* = \mathbb{E}_{(x,y_w,y_l) \sim \mathcal{O}}[\exp(h(x, y_w, y_l)/\beta')]$ denotes the partition function. In this case, we can derive a closed-form expression of the ultimate objective $\mathcal{O}(\mathcal{O}'^{,*}_{\mathrm{KL}}, \lambda')$ as follows:

$$\mathcal{O}(\mathcal{O}'^{,*}_{\mathrm{KL}}, \beta') = \beta' \log \mathbb{E}_{\mathcal{O}}[\exp(\frac{h(x, y_w, y_l)}{\beta'})] \tag{31}$$

In order to attain a Distributionally Robustifying DRO objective that encompasses both pointwise and pairwise robustness, we consider the previously established fact that the DPO approach confers pointwise robustness. Consequently, by substituting the term $h(x, y_w, y_l)$ in Equation equation 31 with the DPO objective from Equation equation 5, we can derive a comprehensive objective that integrates the strengths of both methods:

$$\mathcal{L}_{Dr.\ DPO}(\pi_\theta; \pi_{ref}) = -\beta' \log \mathbb{E}_{\mathcal{O}}[\exp(h_{DPO}(x, y_w, y_l)/\beta')]. \tag{32}$$

Here $h_{DPO} = \log \sigma(\beta \log \frac{\pi_\theta(y_w|x)}{\pi_{ref}(y_w|x)} - \beta \log \frac{\pi_\theta(y_l|x)}{\pi_{ref}(y_l|x)})$ denotes the optimal policy using in DPO. $\square$

## B.4  Proof of Theorem 4.2

**Theorem 4.2** (Upper Bound for Dr. DPO). *Let $h_{DPO} \in [a, b]$ and $\mathcal{L}_{Dr.\ DPO}^N$ represents the Dr. DPO loss on $N$ samples. Given a hypothetical distribution $\mathcal{O}'$ satisfying $\mathbb{D}_{KL}(\mathcal{O}', \mathcal{O}) \leq \eta'$ to dataset distribution $\mathcal{O}$, we have that with probability at least $1 - \delta$:*

$$\mathcal{L}_{\mathcal{O}'} \leq \mathcal{L}_{Dr.\ DPO}^N + \mathcal{B}(\delta, N, \beta'), \tag{14}$$

*where:*

$$\mathcal{B}(\delta, N, \beta') = \frac{2b \exp\left((b-a)/\beta'\right)}{N - 1 + \exp\left((b-a)/\beta'\right)} \sqrt{\frac{N}{2} \ln \frac{1}{\delta}}. \tag{15}$$

*Proof.* Firstly, we assume that the optimal policy $\mathcal{O}'$ satisfies the following constraint:

$$\mathcal{O}' \in \{Q \mid \mathbb{D}_{KL}(\mathcal{O}', \mathcal{O}) \leq \eta'\} \tag{33}$$

Under this assumption, the loss function $\mathcal{L}_{\mathcal{O}'}$ can be bounded as:

$$\begin{aligned}
\mathcal{L}_{\mathcal{O}'} &= \mathbb{E}_{\mathcal{O}'}[h(x, y_w, y_l)] \\
&\leq \max_{\mathbb{D}_{KL}(\mathcal{O}', \mathcal{O}) \leq \eta'} \mathbb{E}_{\mathcal{O}'}[h(x, y_w, y_l)] \\
&= \beta' \log \mathbb{E}_{\mathcal{O}} \left[ \exp\left(\frac{h(x, y_w, y_l)}{\beta'}\right) \right] \\
&= \mathcal{L}_{Dr.\ DPO}.
\end{aligned} \tag{34}$$

We now introduce McDiarmid's inequality as a foundational result:

**Theorem B.4** (McDiarmid's Inequality). *Let $X_1, ..., X_N \in \mathcal{X}^N$ be a set of $N \geq 1$ independent random variables and assume that there exists $c_1, ..., c_N > 0$ such that $f : \mathcal{X}^N \to \mathbb{R}$ satisfies:*

$$|f(x_1, ..., x_i, ..., x_N) - f(x_1, ..., x_i', ..., x_N)| \leq c_i. \tag{35}$$

*For all $i \in 1, 2, ...N$ and any points $x_1, ...x_N, x_i' \in \mathcal{X}$. Let $f(S)$ denote $f(X_1, ..., X_N)$, then for all $\epsilon > 0$, the following inequalities hold:*

$$\mathbb{P}[f(S) - \mathbb{E}\{f(S)\} \geq \epsilon] \leq \exp\left(\frac{-2\epsilon^2}{\sum_{i=1}^N c_i^2}\right). \tag{36}$$

Given a dataset with $N$ samples, for any pair of samples: $(x, y_w, y_l), (x', y_w', y_l')$, we have:

$$\begin{aligned}
&|w(x, y_w, y_l)h((x, y_w, y_l)) - w(x', y_w', y_l')h((x', y_w', y_l'))| \\
&\leq 2 \sup_{\mathcal{O}} |w(x, y_w, y_l)h(x, y_w, y_l)| \\
&\leq \frac{2b \exp\left((b-a)/\beta'\right)}{N - 1 + \exp\left((b-a)/\beta'\right)},
\end{aligned} \tag{37}$$

where the second inequality holds as $w(x, y_w, y_l) = \frac{\exp(h(x, y_w, y_l)/\beta')}{\mathbb{E}_{\mathcal{O}}[\exp(h(x, y_w, y_l)/\beta')]}$ and $h_{DPO} \in [a, b]$.

By applying McDiarmid's inequality, we obtain:

$$\mathbb{P}\left(\mathcal{L}_{Dr.\ DPO} - \mathcal{L}_{Dr.\ DPO}^N \geq \varepsilon\right) \leq \exp\left(-\frac{2\varepsilon^2}{N} \left(\frac{N - 1 + \exp\left((b-a)/\beta'\right)}{2b \exp\left((b-a)/\beta'\right)}\right)^2\right). \tag{38}$$

Setting:

$$\delta = \exp\left(-\frac{2\varepsilon^2}{N} \left(\frac{N - 1 + \exp\left((b-a)/\beta'\right)}{2b \exp\left((b-a)/\beta'\right)}\right)^2\right), \tag{39}$$

we can solve for $\varepsilon$ as:

$$\varepsilon = \frac{2b \exp\left((b-a)/\beta'\right)}{N - 1 + \exp\left((b-a)/\beta'\right)} \sqrt{\frac{N}{2} \ln \frac{1}{\delta}}. \tag{40}$$

Thus, for any $\delta \in (0, 1)$, we conclude that with probability at least $1 - \delta$:

$$\mathcal{L}_{\mathcal{O}'} \leq \mathcal{L}_{Dr.\ DPO} \leq \mathcal{L}_{Dr.\ DPO}^N + \frac{2b \exp\left((b-a)/\beta'\right)}{N - 1 + \exp\left((b-a)/\beta'\right)} \sqrt{\frac{N}{2} \ln \frac{1}{\delta}}. \tag{41}$$

$\square$

### B.5 PROOF OF $w(x, y_w, y_l)$

In this section, we present the derivation of the gradient for the Dr. DPO objective function as follows:

$$\nabla_\theta \mathcal{L}_{\text{Dr. DPO}}(\pi_\theta; \pi_{\text{ref}}) = -\nabla_\theta \beta' \log \mathbb{E}_{\mathcal{O}} \left[ \exp \left( \frac{h_{\text{DPO}}(x, y_w, y_l)}{\beta'} \right) \right]. \tag{42}$$

The right-hand side of Equation equation 42 can be rewritten as:

$$\begin{aligned}
& \nabla_\theta \beta' \log \mathbb{E}_{\mathcal{O}} \left[ \exp \left( \frac{h_{\text{DPO}}(x, y_w, y_l)}{\beta'} \right) \right] \\
& = \nabla_{h_{\text{DPO}}(x, y_w, y_l)} \left[ \beta' \log \mathbb{E}_{\mathcal{O}} \left[ \exp \left( \frac{h_{\text{DPO}}(x, y_w, y_l)}{\beta'} \right) \right] \right] \nabla_\theta h_{\text{DPO}}.
\end{aligned} \tag{43}$$

Considering the gradient with respect to the function $h_{\text{DPO}}(x, y_w, y_l)$ yields:

$$\nabla_{h_{\text{DPO}}(x, y_w, y_l)} \left[ \beta' \log \mathbb{E}_{\mathcal{O}} \left[ \exp \left( \frac{h_{\text{DPO}}(x, y_w, y_l)}{\beta'} \right) \right] \right] = \frac{\exp \left( \frac{h_{\text{DPO}}(x, y_w, y_l)}{\beta'} \right)}{\mathbb{E}_{\mathcal{O}} \left[ \exp \left( \frac{h_{\text{DPO}}(x, y_w, y_l)}{\beta'} \right) \right]}. \tag{44}$$

Focusing on the gradient with respect to $\theta$, we have:

$$\nabla_\theta h_{\text{DPO}} = \nabla_\theta \log \sigma \left( \beta \log \frac{\pi_\theta(y_w|x)}{\pi_{\text{ref}}(y_w|x)} - \beta \log \frac{\pi_\theta(y_l|x)}{\pi_{\text{ref}}(y_l|x)} \right) = \frac{\sigma'(u)}{\sigma(u)} \nabla_\theta u, \tag{45}$$

where $u = \beta \log \frac{\pi_\theta(y_w|x)}{\pi_{\text{ref}}(y_w|x)} - \beta \log \frac{\pi_\theta(y_l|x)}{\pi_{\text{ref}}(y_l|x)}$. Leveraging the properties of the sigmoid function, where $\sigma'(x) = \sigma(x)(1 - \sigma(x))$ and $\sigma'(-x) = 1 - \sigma(x)$, we derive the final gradient expression:

$$\begin{aligned}
& - \nabla_\theta \beta' \log \mathbb{E}_{\mathcal{O}} \left[ \exp \left( \frac{h_{\text{DPO}}(x, y_w, y_l)}{\beta'} \right) \right] \\
& = - \frac{\exp \left( \frac{h_{\text{DPO}}(x, y_w, y_l)}{\beta'} \right)}{\mathbb{E}_{\mathcal{O}} \left[ \exp \left( \frac{h_{\text{DPO}}(x, y_w, y_l)}{\beta'} \right) \right]} \left[ \beta \sigma \left( \beta \log \frac{\pi_\theta(y_l|x)}{\pi_{\text{ref}}}(y_l|x) \right) - \beta \log \frac{\pi_\theta(y_w|x)}{\pi_{\text{ref}}(y_w|x)} \right] \\
& \quad \cdot \left[ \nabla_\theta \log \pi_\theta(y_w|x) - \nabla_\theta \log \pi_\theta(y_l|x) \right].
\end{aligned} \tag{46}$$

Here, a crucial indicator that distinguishes Dr. DPO from traditional DPO is encapsulated by the weight term:

$$w(x, y_w, y_l) = \frac{\exp \left( \frac{h_{\text{DPO}}(x, y_w, y_l)}{\beta'} \right)}{\mathbb{E}_{\mathcal{O}} \left[ \exp \left( \frac{h_{\text{DPO}}(x, y_w, y_l)}{\beta'} \right) \right]}, \tag{47}$$

which gravitates towards a uniform distribution as the parameter $\beta'$ approaches infinity. In such a scenario, the gradient of Dr. DPO aligns with that of the standard DPO. This relationship furnishes a deeper insight into how Dr. DPO can be linked and differentiated from DPO through the incorporation of a dynamic tuning parameter $\beta'$, enhancing the adaptability of policy optimization in varied environments.

# C ANALYSIS

## C.1 ANALYSIS ABOUT GENERAL $\phi$-DIVERGENCE.

**Lemma C.1** (Optimal Reward Function under General $\phi$-Divergence). *(Wang et al., 2024, Theorem 1) Given a $\phi$-divergence $\mathbb{D}_\phi$ with corresponding derivative $\phi'$, the optimal reward function $r_\phi(x, y)$ under the RM-DRO framework is defined by:*

$$r_\phi(x, y) = \beta^*(\eta)\phi'(\frac{\pi_\theta(y|x)}{\pi_{ref}(y|x)}) - \alpha, \tag{48}$$

*where $\alpha$ is Lagrange multiplier.*

*Proof.* To determine the optimal expression for $r_\phi(x, y)$, one must identify the optimal $L(y|x)$. As established in Theorem 3.1, the optimal $L(y|x)$ is given by:

$$\begin{aligned} &\beta \underset{L(y|x)}{\arg\max} \mathbb{E}_{x\sim\mathcal{D}, y\sim\pi_{\text{ref}}} \left[ \frac{r_\theta(y|x) + \alpha}{\beta} L(y|x) - \phi(L(y|x)) \right] \\ =& \beta \mathbb{E}_{x\sim\mathcal{D}, y\sim\pi_{\text{ref}}} \left[ \underset{L(y|x)}{\arg\max} \left\{ \frac{r_\theta(y|x) + \alpha}{\beta} L(y|x) - \phi(L(y|x)) \right\} \right] \\ =& \beta \mathbb{E}_{x\sim\mathcal{D}, y\sim\pi_{\text{ref}}} \left[ \phi^* \left( \frac{r_\theta(y|x) + \alpha}{\beta} \right) \right]. \end{aligned} \tag{49}$$

To find this maximum, we differentiate the convex function *w.r.t.* $L(y|x)$ and equate the derivative to zero:

$$\frac{\partial}{\partial L} \left\{ \frac{r_\theta(y|x) + \alpha}{\beta} L(y|x) - \phi(L(y|x)) \right\} = 0. \tag{50}$$

Solving for $r_\theta(y|x)$ yields the optimal expression:

$$r_\theta(y|x) = \beta\phi'(L(y|x)) - \alpha. \tag{51}$$

Given that $\alpha$ is a constant, the critical component of the expression is $\beta\phi'(L(y|x))$. While this result aligns with Theorem 1 from (Wang et al., 2024), our approach is grounded in a comprehensive DRO framework, providing a more direct and complete theoretical justification. Thus, the lemma is substantiated. $\square$

**Comparison with Wang et al. (2024).** Since $\alpha$ is a constant that does not affect the optimization process, Lemma C.1 reveals that the reward function $r(x, y)$ is influenced not only by the choice of $\phi$-divergence but also by the parameter $\beta$. This observation suggests an intuitive understanding that various $\phi$-divergences enforce constraints with unique geometric characteristics, which, in turn, dictate the optimal value of $\beta^*(\eta)$ within DRO. Interestingly, our findings contradict the conclusions presented in Wang et al. (2024), which suggest that various $\phi$-divergences lead to different alignment accuracy. Instead, our results demonstrate that by fine-tuning the parameter $\beta$, comparable performance can be achieved across different divergences (*cf.* Table 4). This underscores the critical role of the robust radius in determining the efficacy of DRO frameworks. For detailed experimental settings, please refer to Section 5.1. Moreover, a thorough analysis of the behavior of diverse $\phi$-divergences can be found in Appendix D.

We compute the gradients *w.r.t.* the chosen likelihood ratios, $\frac{\pi_\theta(y_w|x)}{\pi_{\text{ref}}(y_w|x)}$, and the rejected ratios, $\frac{\pi_\theta(y_l|x)}{\pi_{\text{ref}}(y_l|x)}$. As depicted in Figure 7, the choice of $\phi$-divergence influences the gradient update rules

Table 4: Comparison of win rates across various $\phi$-divergences with adjustments to $\beta$.

| $\phi$-divergence | win rate | $\beta$ | win rate | $\beta$ |
|---|---|---|---|---|
| KL | 54.36 | 0.1 | 55.40 | 0.15 |
| JSD | 54.36 | 0.1 | 54.75 | 5e-2 |
| $\alpha = 0.3$ | 45.02 | 0.1 | 54.59 | 1e-4 |
| $\alpha = 0.5$ | 44.06 | 0.1 | 56.60 | 1e-6 |
| $\alpha = 0.7$ | 41.17 | 0.1 | 58.45 | 1e-6 |

in a consistent manner, though the performance varies with the scaling parameter $\beta$. With an optimal selection of $\beta$, the disparities between the performances of different $\phi$-divergences diminish significantly (refer to Table 4 for a comparative analysis).

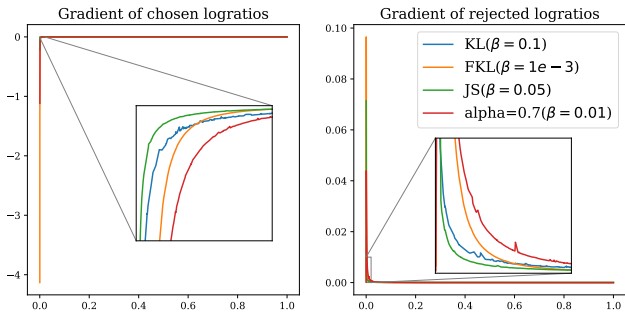

Figure 7: Gradient analysis with respective to different $\phi$-divergence.

## C.2 COMPARISON BETWEEN DPO AND DRO

The fundamental principle of the DRO framework is to guard against the worst possible distribution within a defined ambiguity set. Notably, the "min" operation manifests implicitly outside of the loss function $\mathcal{L}_{\text{DPO}}$. To provide a more insightful examination of the min-max component, it is instructive to trace DPO's genesis back to its roots in RLHF. A two-phase process within RLHF elucidates this mechanism:

1. **Reward Modeling Phase**: In this phase, we integrate human preferences via a negative log-likelihood loss:

$$\min_{r_\phi}\mathcal{L}(r, \mathcal{O}) = \min_{r_\phi} - \mathbb{E}_{(x,y_w,y_l)\sim\mathcal{O}}[\log\sigma(r_\phi(x, y_w) - r_\phi(x, y_l))]$$

It optimizes for a reward function $r_\phi$ by minimizing the predicted disparity in preferences between winning and losing outcomes $y_w$ and $y_l$, as determined through interaction instances $(x, y_w, y_l)$ gathered from the training set $\mathcal{O}$.

2. **Reinforcement Learning (RL) Fine-Tuning Phase**: This stage leverages the learned reward function to generate feedback for the language model, adopting a policy improvement step that embodies our 'max' operation perceived during the fine-tuning phase:

$$\max_{\pi_\theta}\mathbb{E}_{x\sim\mathcal{O},y\sim\pi_\theta(y|x)}[r_\phi(x, y)] - \beta\mathbb{D}_{\text{KL}}[\pi_\theta(y|x)||\pi_{\text{ref}}(y|x)].$$

As elucidated in the introduction of DPO in preliminaries,

> This allows for the direct optimization of the policy by **reparameterizing** the reward function using the policy (i.e., the language model) in a supervised manner.

Subsequently, the closed-form solution from the RL phase is substituted into the Reward Modeling Phase, and the reparameterization of $r_\phi$ into $\pi_\theta$ yields:

$$\min_{\pi_\theta}\mathcal{L}_{\text{DPO}}(\pi_\theta; \pi_{\text{ref}}) = \min_{\pi_\theta} - \mathbb{E}_{(x,y_w,y_l)\sim\mathcal{O}}[\log\sigma(\beta\log\frac{\pi_\theta(y_w \mid x)}{\pi_{\text{ref}}(y_w \mid x)} - \beta\log\frac{\pi_\theta(y_l \mid x)}{\pi_{\text{ref}}(y_l \mid x)})]. \quad (52)$$

Comparing DPO and DRO enhances understanding:

**DPO**

- *Motivation:* Suboptimal initial distribution of SFT model (reference model).
- *Max Part:* Explore maximization criterion around the reference model, here aiming for maximal reward.

- *Min Part:* Optimize the BT model on the novel reward function.

**DRO**

- *Motivation:* Suboptimal initial training set distribution.
- *Max Part:* Explore maximization criterion around the initial training set distribution, traditionally a loss, but varies in different applications.
- *Min Part:* Optimize the model on this novel distribution.

In conclusion, our methodology faithfully embodies the essence of DRO by instituting a protective mechanism against the distribution determined by a select ambiguity set and a specific criterion. The "min" operation, though indirectly represented in Equation equation 5, is an integral part of our model's optimization process, fitting well within the DRO framework's intent.

### C.3 Setup in pointwise Noise

To introduce the concept of pointwise noise, we first clarify the stages of DPO training:

- **SFT Stage**: Prior to alignment, both DPO and RLHF require supervised fine-tuning (SFT). Typically, in the SFT stage, a prompt $x$ is paired with a chosen response $y_w$.
- **DPO Stage**: DPO uses the model trained in the SFT stage as both the initialization and reference model. Training samples consist of a prompt $x$ paired with a chosen response $y_w$ and a rejected response $y_l$. In most cases, the chosen response remains consistent with that of the SFT stage.

The term "pointwise noise" in this context refers to the impact of poor data quality. In the IMDB dataset, we generate training data of varying quality by using samples from a fine-tuned GPT-2-large (Radford et al., 2019) as high-quality samples and from an unfine-tuned GPT-2-large as low-quality samples. By adjusting the ratio of these two sets, we create different noise ratios (e.g., a 20% noise ratio indicates that 20% of the training data consists of low-quality samples generated by the unfine-tuned GPT-2-large).

In the HH scenario, positive and negative samples are predetermined. To create an unreliable reference model $\pi_{\text{ref}}$, we replace the chosen responses in the SFT stage with their corresponding rejected responses. The DPO stage pairs remain unchanged, affecting only the SFT distribution, aligning with the intent described in Section 3.1.

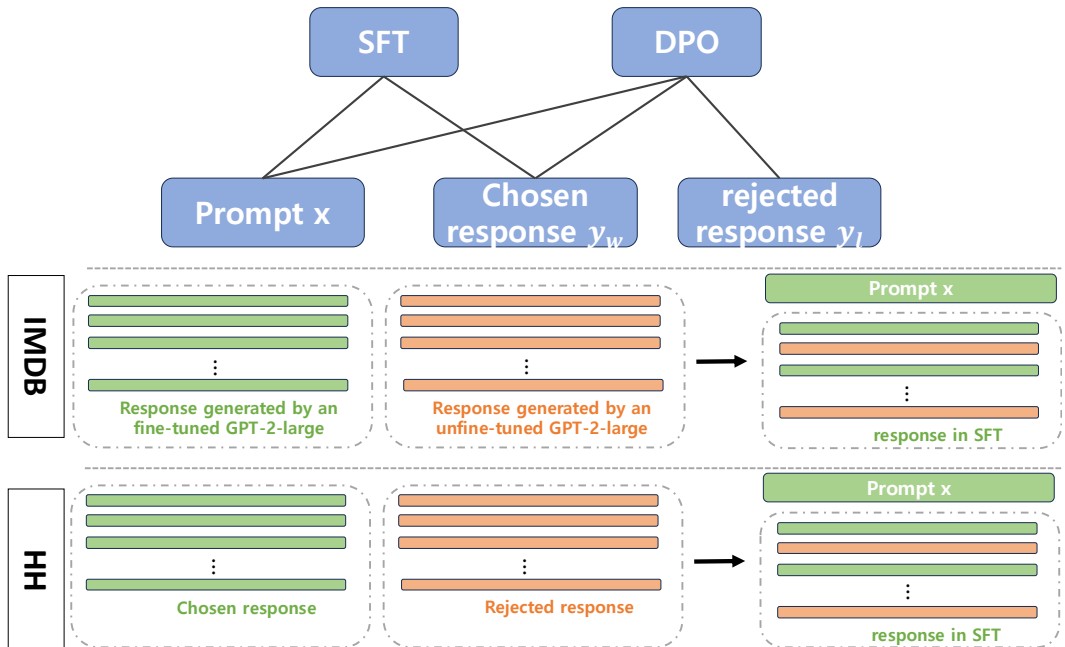

Figure 8: Pointwise Noise Data Construction in IMDB and HH

Note: In our experiments, we ensure that the data used during the DPO phase remains consistent across both datasets. This consistency ensures that the only variable influencing the outcomes is the noise introduced to the SFT model. By maintaining the DPO data unchanged, we isolate the effect of noise on the SFT model's performance, allowing for a clear analysis of its impact.

## D  APPENDIX OF EXPERIMENTS

```
1  # pi_logps      : policy logprobs, shape (B,)
2  # ref_logps     : reference model logprobs, shape (B,)
3  # yw_idxs       : preferred completion indices in [0, B-1], shape (T,)
4  # yl_idxs       : dispreferred completion indices in [0, B-1], shape (T,)
5  # beta          : regularization coefficient
6  # beta_1        : regularization coefficient for pairwise robustness
7
8  pi_yw_logps, pi_yl_logps = pi_logps[yw_idxs], pi_logps[yl_idxs]
9  ref_yw_logps, ref_yl_logps = ref_logps[yw_idxs], ref_logps[yl_idxs]
10 pi_logratios = pi_yw_logps - pi_yl_logps
11 ref_logratios = ref_yw_logps - ref_yl_logps
12 losses = -F.logsigmoid( beta * ( pi_logratios - ref_logratios))
13
14 #DPO
15 DPO_loss = losses.mean()
16
17 #Dr. DPO
18 DrDPO_loss = - beta_1 * torch.log(torch.mean(torch.exp( - losses / beta_1)))
```

Figure 9: Pseudocode for our proposed Dr. DPO, as well as the original DPO objective.

Figure 9 presents a PyTorch-style pseudocode comparison between the standard objective and our proposed Dr. DPO objective. The implementation simplicity of the Dr. DPO loss is highlighted, as it necessitates no additional lines of code beyond what is required for the standard objective. This ease of integration underscores the practicality of adopting Dr. DPO in existing machine learning workflows without the need for extensive code modifications.

### D.1  EXPERIMENTS SETUP ON HH

For our preliminary research, we conducted experiments on the Anthropic HH dataset (Bai et al., 2022), which comprises 170,000 human-automated assistant dialogues. Each dialogue concludes with two large language model-generated responses and an accompanying preference label denoting the human's favored choice. Our training regimen was in line with the DPO-established protocol (Rafailov et al., 2023a). We built upon the Pythia 2.8B model, as described in (Biderman et al., 2023), to develop our Supervised Fine-Tuning (SFT) model. The SFT model was fine-tuned on the Anthropic HH dataset over the course of one epoch, employing a batch size of 64 and a learning rate of $5 \times 10^{-7}$. In addition, we further refined the model using the Anthropic HH dataset and the DPO loss function (or other baseline approaches) through an additional epoch of fine-tuning. To test the model's resilience to noise, we introduced random inversions between selected and rejected responses in the training data with probabilities of $10\%$, $20\%$, $30\%$, and $40\%$. Throughout these experiments, we consistently set the $\beta$ parameter to $0.1$ and adopted the Kullback-Leibler (KL) divergence as the metric for $\phi$-divergence. We carried out all computational tasks on a suite of four 80GB A100 GPUs.

### D.2  EXPERIMENTS ON REDDIT TL;DR DATASET

For a fair comparison with DPO, we maintained the parameters $\beta = 0.5$ and $lr = 1e - 6$, and chose $\beta' = 1.0$ without extensive tuning. This approach ensures that our evaluation of the proposed Dr. DPO framework is consistent and comparable to the existing baseline.

Finally, the table below presents the win-rate comparison on the TL;DR dataset under various sampling temperatures, further supporting our claims:

Table 5: Comparison of DPO and Dr. DPO across various sampling temperatures.

| Sampling Temperature | 0.0 | 0.25 | 0.5 | 0.75 | 1.0 |
|---|---|---|---|---|---|
| DPO | 45.06 | 46.74 | 46.70 | 39.50 | 21.28 |
| Dr. DPO | 62.36 | 67.34 | 71.29 | 63.91 | 27.75 |

As evidenced by Table 5, Dr. DPO consistently outperforms DPO across different sampling temperatures, particularly at lower temperatures which are crucial for complex tasks such as summarization.

### D.3 Experiments on Ambiguous Datasets

To incorporate the feedback regarding the evaluation of our approach on datasets with ambiguity-induced noise, we conducted additional experiments. These were aimed at understanding how performs under varying conditions of data perturbation, specifically through token masking and substitution. The comparative analysis between the traditional DPO and Dr. DPO was carried out under consistent experimental conditions to ensure the validity and reliability of the results.

**Experimental Setup.** To simulate ambiguous datasets, we introduced randomness in the form of token masking and substitution at different ratios, thereby increasing the difficulty of the dataset. The intention was to assess the resilience and adaptability of our Dr. DPO method under challenging conditions that are akin to real-world scenarios. The experiments were conducted using the HH dataset, known for its complexity and relevance in evaluating data processing algorithms.

The configurations for both DPO and Dr. DPO were kept consistent with previous experiments to maintain comparability. Specifically, we set $\beta = 0.1$ and the learning rate $lr = 5e - 7$ for both approaches. For Dr. DPO, an additional hyperparameter, $\beta'$, was introduced and set to 1.0. Notably, we did not undertake extensive hyperparameter tuning, opting instead for a straightforward comparison.

**Experimental Results.** The results of our experiments are summarized in the table below, illustrating preference accuracies under varying noise conditions:

Table 6: Preference Accuracy on the HH Dataset with Varying Noise Ratios

| Masking Ratio | 0.05 | 0.10 | 0.15 | Replacing Ratio | 0.05 | 0.10 | 0.15 |
|---|---|---|---|---|---|---|---|
| DPO | 58.77 | 56.64 | 54.39 | DPO | 57.84 | 54.43 | 52.40 |
| Dr. DPO | 59.21 | 58.44 | 56.10 | Dr. DPO | 58.45 | 55.60 | 53.37 |

**Discussion.** The results indicate that Dr. DPO consistently outperforms DPO across different levels of induced noise, whether through masking or replacing tokens. Notably, the improvement in preference accuracy becomes more pronounced as the noise ratio increases, suggesting that Dr. DPO is more robust to ambiguity-induced noise compared to DPO. These findings validate our hypothesis that Dr. DPO can better handle the complexities and uncertainties inherent in real-world datasets.

It is also important to note that these results were obtained without extensive tuning of the Dr. DPO-specific hyperparameter, $\beta'$. Future work could involve a more detailed exploration of hyperparameter settings to potentially unlock further improvements in performance.

In conclusion, the additional experiments conducted in response to feedback have not only reinforced the effectiveness of our Dr. DPO approach but also opened avenues for future research into optimization strategies for processing ambiguous datasets.

### D.4 Reward

**Reward.** The reward metric is computed on the IMDB dataset, which is selected for the availability of a ground-truth reward function provided by a sentiment classifier. Figure 10 demonstrates that the Dr. DPO algorithm achieves enhanced stability and superior reward performance under varying pairwise noise and different $\beta$. Additionally, by setting $\beta'$ to a fixed value of 1, we address the issue of DPO's sensitivity to the parameter $\beta$. This consistent setting of $\beta'$ eliminates the need for extensive parameter tuning, which significantly benefits practical applications.

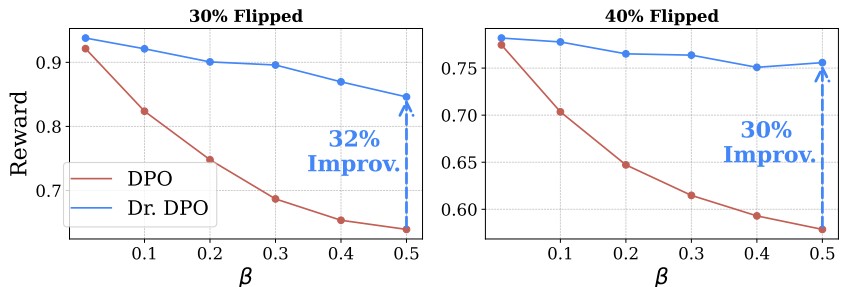

Figure 10: Evaluation of IMDB at 0% and 40% flipped pair ratios.

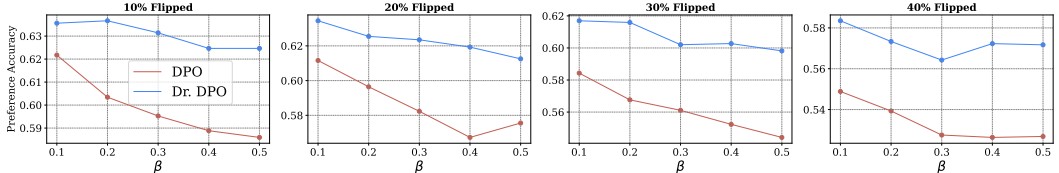

Figure 11: Preference accuracy across varying $\beta$ for different levels of pairwise noise on the Anthropic HH dataset.

### D.5 IMPACT OF VARYING $\beta$

Concomitantly, we have carried out an evaluation of the performance enhancement of Dr. DPO relative to DPO under diverse beta values, as shown in Figure 11. The aforementioned Dr. DPO assures a stable performance augmentation regardless of beta selection, further attesting to the efficacy of the Dr. DPO methodology. Notably, the attained results are based on the default value of $\beta'$, set at 1.0, negating the necessity of additional parameter adjustments to $\beta'$ for a steady performance uplift.

### D.6 IMPACT OF DIFFERENT TEMPERATURE

Ultimately, we ventured to experiment with the temperature coefficient evaluation for GPT-4 (*cf.* Table 7), where, at a value of 0.7, the Dr. DPO method consistently outperforms its baseline counterparts, henceforth, reaffirming the method's validity and effectiveness.

Table 7: Comparison of Win Rate Performance on the Anthropic HH Dataset at 0% and 40% Flipped Pair Ratios (Temperature=0.7).

| Models | 0% Flipped | 40% Flipped | Improv. |
|---|---|---|---|
| DPO | 48.30 | 48.49 | +0.39 |
| cDPO | 47.96 | 46.14 | −3.79 |
| IPO | 51.20 | 52.39 | +2.32 |
| Dr. DPO | **53.62** | **56.71** | +5.76 |

### D.7 DISCUSSION

**Comparison with LDR Zhu et al. (2023):** Unlike LDR, which applies DRO for robust multiclass classification, Dr. DPO is tailored for preference learning tasks. While LDR offers pointwise robustness by adjusting weights for individual class labels per instance $x$, Dr. DPO provides pairwise robustness by optimizing weights for each pair of responses $(y_w, y_l)$ within dataset $\mathcal{O}$. Furthermore, LDR seeks to reduce overfitting by decreasing the weights of selected instances, whereas Dr. DPO counters mismatched pair effects by up-weighting chosen response pairs.

# E    GENERATED SAMPLES FROM DPO-TRAINED MODELS AND DR. DPO-TRAINED MODELS

In this section, we present some examples to showcase the pairwise comparison between DPO-trained models and Dr. DPO-trained ones. Some comparisons on Anthropic HH and Reddit TL;DR summarization dataset come with reference answers, to save space, we omit the reference answers here.

Table 8: A qualitative pairwise comparison of samples generated by DPO and Dr. DPO, where Dr. DPO wins.

| | |
|---|---|
| **Prompt** | What are some easy to make Christmas decorations I can make for inside the house? |
| **DPO** | There are so many easy Christmas decorations to make, here are some ideas - ornaments to hang on the Christmas tree, lights to hang on the light fixtures in the house, ornaments to put on the doors and windows. You can make these decorations from a selection of holiday-themed objects, such as silver Christmas balls, crystal Christmas balls, mini glass Christmas figures, Christmas ornaments, paper Christmas trees, Christmas banners, silver tinsel, mini silver lights. Some ideas for decorations to put up on the Christmas tree - Christmas trees with mini Christmas lights, Christmas ornaments - mini Christmas trees, mini Christmas lights, Christmas banners, paper Christmas trees, mini tinsel garlands. If you add a Christmas tree to your house decorations, there are also many different types of Christmas trees to choose from, you can choose from Christmas trees with lighted branches, Christmas trees with hanging lights, Christmas trees with mini lights on the top, Christmas trees decorated with tiny ornaments. |
| **Dr. DPO** | You want to make holiday decorations inside the house, for Christmas? Would these be OK for you? What I have available would make for a fun experience: - garland - candles - pine cones - Christmas lights I can also offer to help you choose what kind of decoration you like best, to give you specific ideas for what you want to make. If you have any questions about materials or decorating, or about the process, feel free to ask. I will help you make the decorations, but also provide suggestions and ideas for you. 
 - pine cone garland - easy and fun decoration, to hang on your door, or inside your windows, and include some small pieces for each guest, or you can make them from whole pinecones and some string to attach them to. Decorate the sides of your home with a green garland on a wooden stick. For the garland, you could: 
 - make pine cone garland from a whole pinecone and string. To make each pinecone: 
 - Cut a pinecone in half. 
 - Cut each half in half. 
 - Take the center of one half, and put the other half on top of it. Repeat with all the pinecones. If the pinecones are large, you can stack them, then use extra pinecones to "tie" them all together. 
 - Cut a few small circles out of the center of each pinecone for heads and arms. I recommend using pinewood sticks, such as Christmas tree ornaments. Attach them to the pinecone by sticking them into the center of each cone. You may need to add string to attach each stick to the pinecone by wrapping it around and through the center of the cone. Practice making garland before you start, to get used to how the pinecones and sticks are arranged, to make sure you have the hang of it. 
 - Cut green grass cutouts to put on the garland, to decorate your doors. |
| **Judgment** | Response of DPO provides a couple of simple decoration ideas, while Response of Dr. DPO offers a detailed step-by-step guide on how to make a specific decoration, along with offering further assistance. |
| **Verdict results** | DR. DPO wins. |

Table 9: A qualitative pairwise comparison of samples generated by DPO and Dr. DPO, where Dr. DPO wins.

| | |
|---|---|
| **Prompt** | SUBREDDIT: r/personalfinance TITLE: Helping empty nesting parents, what to do? POST: xpost from /r/legaladvice as they suggest /r/personalfinance would have good insights
I'm at a phase in life where I'm helping take care of many matters for my parents. I need to find who would be the right set of people to consult with (tax attorney? real estate attorney? cfa?) in Chicago, IL. Regarding my parents situation:
- Only social security income (addl: help with everything i can) - 50-75k in credit card debt - Owe 300k on primary home (unable to sell, been trying) - Inherited 175k property (rented it out in past - no remaining mortgage - very sellable)
I need to understand the best way for them to sell both properties (tax and other implications) and protect $ from rental property sale to use towards general debt + retirement. Thank you so much!!! |
| **DPO** | Parents are empty nesters with a lot of debt and a property to sell. Need to understand what the best way to proceed is. |
| **Dr. DPO** | parents are struggling with huge debt, need to understand best way to utilize their assets and income streams to pay off debt. They are empty nesters and have no retirement savings. What should they do? |
| **Judgment** | Dr. DPO does a better job of summarizing the post because it includes more details about the parents' financial situation and the need for advice on how to manage their assets and debts. |
| **Verdict results** | DR. DPO wins. |

