# OpenReview forum: "Towards Robust Alignment of Language Models: Distributionally Robustifying Direct Preference Optimization"
_ICLR.cc/2025/Conference — ICLR 2025 Poster_

### Official Review · Reviewer_SaBc · 2024-10-29

**Soundness:** 3
**Presentation:** 3
**Contribution:** 3
**Rating:** 6
**Confidence:** 4

**Summary:**

The paper examines the robustness of Direct Preference Optimization (DPO) in aligning large language models (LLMs) with human preferences and introduces "Distributionally Robustifying DPO" (Dr. DPO) as a framework to enhance this robustness. The authors identify two types of noise in training data: pointwise noise (low-quality data in supervised fine-tuning) and pairwise noise (incorrect preference rankings in preference training). For pointwise noise, they show that the original DPO framework inherently applies DRO principles, where the parameter $\beta$ correlates with the robustness radius $\eta$ in the DRO framework, reflecting tolerance to pointwise errors in the data. To address pairwise noise, they extend DPO with Dr. DPO by introducing a new hyperparameter that adjusts the impact of noisy pairs, resulting in improved performance in challenging, noisy environments. Empirical results validate Dr. DPO’s effectiveness in producing high-quality responses and sustaining accuracy across various noise levels.

**Strengths:**

Originality:
- The idea of applying Distributionally Robust Optimization to LLM alignment is novel and interesting.

Clarity:
- The paper is well writen and easy to follow.

Significance:
- The proposed Dr. DPO method has natual interpretation and easy to compute.
- Experimental results are promising. Auther compres the performance of Dr.DPO with various alignment methods and provide detailed ablation studies and parameter $\beta'$.

**Weaknesses:**

Major concerns:
- The discussion on pointwise noise and DRO lacks clarity. Equation 7 appears to be a dual form of the PPO objective (Equation 3). Typically, DRO aims to optimize a target function over distributions close to the empirical sample distribution. In reward modeling, the "sample" refers to the input prompt and the answer generated by the target model, $\pi_{\theta}$. Following this logic, the DRO objective would optimize over distributions $Q$ close to $(x, \pi_{\theta}(y|x))$. However, in Equation 7, the authors approach the problem as optimizing objectives over distributions $\pi_{\theta}(\cdot|x)$ near $\pi_{ref}(\cdot|x)$. I don't see a clear connection between equaiton 7 and DRO. Providing an interpretation of Equation 7 and discussing its relation to Equation 6 would help clarify this connection.


Minor concerns:
- The model used in pairwise experiments is not explicitly stated in the main text. The Pythia 2.8B model is mentioned only in the appendix, which reduces confidence in the experimental results.
- The authors have omitted relevant literature on applying Distributionally Robust Optimization to LLMs, specifically, Oren, Yonatan, et al.'s "Distributionally Robust Language Modeling."
- Setting $\beta' = 1$ results in $L_{Dr.DPO} = -\log E[\exp(h_{DPO})]$, which differs slightly from the original DPO formula. A more detailed interpretation of this form would be insightful.
- In Figure 5 (left), Dr. DPO shows higher win and loss rates at 0 flips. Does this imply that incorporating Dr. DPO leads to a trade-off between performance and robustness?

**Questions:**

I look forward to discussing these questions further in the weaknesses section, as they have the potential to either increase or decrease the score.

---

> ### Author Response · Authors · 2024-11-22
>
> We sincerely thank you for your valuable feedback and constructive comments. Below, we address each of your concerns in detail.
>
> ---
> **Q1: The discussion on pointwise noise and DRO lacks clarity.**
>
> **A1:** We appreciate the opportunity to clarify. **The key point is that DPO fundamentally operates as an offline algorithm, where both DPO and Dr. DPO rely on pairwise data sampled from $\pi_{\text{ref}}$ (or $\pi_{\text{SFT}}$).** Let us explain the sample acquisition process, particularly using datasets like "princeton-nlp/llama3-ultrafeedback":
>
> 1. For each prompt $x$, the SFT model generates five responses with a sampling temperature of 0.8.
> 2. These responses are scored using "llm-blender/PairRM," with the highest-scoring response selected as $y_w$ (winner) and the lowest as $y_l$ (loser).
>
> Thus, the empirical sample distribution is inherently tied to $\pi_{\text{SFT}}$, and DPO consistently operates under this static distribution.
>
> In contrast, PPO is an online algorithm. It dynamically updates the policy $\pi_\theta$ during training, requiring the generation of new samples aligned with the current $\pi_\theta$. This fundamental difference means that PPO explicitly interacts with its evolving policy, while DPO remains constrained to the fixed $\pi_{\text{SFT}}$.
>
> To further elucidate, we have added a detailed connection between DRO and DPO in `Appendix C.2`, highlighting their shared optimization framework and differing motivations.
>
> | Method | Motivation | Max Part | Min Part |
> |----------|----------------|----------|----------|
> | DRO | Suboptimal initial training set distribution | Explore maximization criterion around the initial training set distribution, traditionally a loss, but varies in different applications. | Optimize the model on this novel distribution |
> | DPO | Suboptimal initial distribution of SFT model (reference model)| Explore maximization criterion around reference model, here aiming for maximal reward | Optimize BT model on the novel reward function|
>
> **Q2: The model used in pairwise experiments is not explicitly stated in the main text. The Pythia 2.8B model is mentioned only in the appendix, which reduces confidence in the experimental results.**
>
> **A2:** Thank you for pointing this out. To address this concern, we have explicitly stated in the main text that all pairwise experiments are conducted on the Pythia 2.8B model unless a different base model is explicitly mentioned. This follows the experimental setup outlined in the original DPO work.
>
> **Q3: The authors have omitted relevant literature on applying Distributionally Robust Optimization to LLMs, specifically, Oren, Yonatan, et al.'s "Distributionally Robust Language Modeling."**
>
> **A3:**  We appreciate the suggestion to include this important reference. We have incorporated a discussion of "Distributionally Robust Language Modeling" by Oren et al. into the related work section. This reference highlights the broader application of DRO principles to language modeling, providing additional context for our contributions.

---

> > ### Comment · Reviewer_SaBc · 2024-11-25
> > **Q2,Q3 solved but still have concerns with Q1**
> >
> > Thanks for the clearify by the authors, the sample acquisition process is helpful for understanding. I under stand DPO is an offline algorithms but stil have concerns. The main goal of DRO is to tackle the uncertainty in training data. Let $L(x, \theta)$ be the original loss function, and $Q(x)$ be the empirical distribution of $x$, the DRO objective should be
> > \begin{align}
> > L_{DRO}(\theta) = max_{Q'} E_{x \sim Q'(x)}[L(x, \theta)] ~~ s.t. \mathcal{D}(Q',Q) \leq \eta
> > \end{align}
> > and the parameter is estimed by
> > \begin{align*}
> > \hat{\theta} = \arg\min_{\theta} L_{DRO}(\theta).
> > \end{align*}
> > But when it comes to equation 7, I cannot see how it is related to DRO formulation. If $\pi_{ref}$ or $\pi_{SFT}$ is the empirical distribution (relate to $Q$ in the above equation), then $\pi_{\theta}$ is $Q'$? In DRO $Q'$ should be arbitary distribution around $Q$ rather than a parameterized distribution. DRO contains two optimization process, the estimation of $\theta$ should be conducted in the second equation above, after obtaining $L_{DRO}(\theta)$ by maximizing the loss around distributions around empirical distribution. Equation 7 seems to mess things up.
> >
> > Therefore, I hold my view that the authors have a misunderstanding of DRO in pointwise noise section and equation 7 is an optimization with KL regularization not a DRO.
> >
> > If the authors can point out what how the terms their equation 7 is related to $L_{DRO}(\theta)$, $Q$, $Q'$ in the equations above, I would love to change my mind.

---

> ### Author Response · Authors · 2024-11-22
>
> **Q4: Setting $\beta' = 1$results in $L\_{\text{DR.DPO}} = -\log \mathbb{E}[\exp(h\_{\text{DPO}})]$, which differs slightly from the original DPO formula. A more detailed interpretation of this form would be insightful.**
>
> **A4:** The primary difference between the original DPO formula and DrDPO lies in the introduction of the `Log-Expectation-Exp` structure:
> $$
> L\_{\text{DR.DPO}} = -\beta' \log \mathbb{E}\_{\mathcal{O}} \left[\exp\left(\frac{h\_{\text{DPO}}(x, y\_w, y\_l)}{\beta'}\right)\right].
> $$
>
> This structure offers advantages for handling label-flipped noise. Examining the gradient formulation provides further insight:
> $$
> \frac{\partial}{\partial h\_{\text{DPO}}(x, y\_w, y\_l)} \left(-\beta' \log \mathbb{E}\_{\mathcal{O}} \left[\exp\left(\frac{h\_{\text{DPO}}(x, y\_w, y\_l)}{\beta'}\right)\right]\right) = -\frac{\exp\left(\frac{h\_{\text{DPO}}(x, y\_w, y\_l)}{\beta'}\right)}{\mathbb{E}\_{\mathcal{O}} \left[\exp\left(\frac{h\_{\text{DPO}}(x, y\_w, y\_l)}{\beta'}\right)\right]}.
> $$
>
> This reweighting mechanism assigns higher weights to samples with larger $h\_{\text{DPO}}(x, y\_w, y\_l)$ when $\beta'$ is small. As $\beta'$ grows large, the gradient aligns with the original DPO formulation, reverting to a uniform distribution.
>
> We empirically found $\beta' = 1$ to be a well-performing hyperparameter for our experiments.
>
>
> **Q5: In Figure 5 (left), Dr. DPO shows higher win and loss rates at 0 flips. Does this imply that incorporating Dr. DPO leads to a trade-off between performance and robustness?**
>
> **A5:** The 0% flipped case indicates that we did not introduce any label flips to the dataset, i.e., the original dataset. However, this does not imply that the dataset is devoid of label noise. Dr.DPO continues to outperform DPO and IPO in the 0% flipped case scenario, further substantiating the presence of some label flipping noise within existing datasets, which also serves as the motivation for Dr.DPO. This finding aligns with the results of rDPO [1], where a default flip rate of 0.1 was shown to achieve better performance on the HH dataset, as evidenced in Table 3 of their work.
>
>
> **References:**
> [1] Chowdhury, S. R., Kini, A., & Natarajan, N. (2024). Provably robust DPO: Aligning language models with noisy feedback. *ICML 2024*.
>
> We hope these clarifications address your concerns comprehensively. Thank you once again for your constructive feedback and the opportunity to improve our work.

---

> ### Author Response · Authors · 2024-11-26
>
> We sincerely thank you for your valuable feedback and constructive comments. Below, we address each of your concerns in detail.
>
> ---
> **Q6: If $\pi\_{\text{ref}}$ or $\pi\_{\text{SFT}}$ is the empirical distribution (relate to $Q$ in the above equation), then is $\pi\_{\theta}$ $Q'$?**
>
> **A6:** Your observation is correct that Equation (7) might lead to some misunderstanding. For a more precise formulation, the problem can be expressed as:
> $$\max\_{\pi\_r} \mathbb{E}\_{x \sim \mathcal{O}, y \sim \pi\_r(y|x)}[r\_\phi(x, y)] \quad \text{s.t.} \mathbb{D}\_\phi(\pi\_r(y|x), \pi\_{\text{ref}}(y|x)) \leq \eta.$$
> In this context:
> - $\pi\_{\text{ref}}$ (or $\pi\_{\text{SFT}}$) acts as the base empirical distribution, analogous to $Q$ in classical DRO.
> - $\pi\_r$ serves as $Q'$, the alternative distribution we optimize over to identify the optimal reward function under the ambiguity constraint.
>
> **Q7: "The estimation of $\theta$ should be conducted in the second equation above, after obtaining $L\_{\text{DRO}}(\theta)$ by maximizing the loss around distributions around the empirical distribution."**
>
> **A7**: The standard DRO framework typically consists of two optimization processes:
> 1. **Inner Maximization**:
>    Identify the distribution $Q'$ within the ambiguity set, which maximizes the expected loss:
>    $$Q' = \arg\max\_{Q': \mathcal{D}(Q', Q) \leq \eta} \mathbb{E}\_{x \sim Q'}[L(x, \theta)].$$
> 2. **Outer Minimization**:
>    Minimize the obtained loss with respect to the model parameters $\theta$:
>    $$\hat{\theta} = \arg\min\_{\theta} \mathbb{E}\_{x \sim Q'}[L(x, \theta)].$$
>
> **In our method, we achieve an analogous structure but with a focus on the preference-based framework**:
>
> 1. **Inner Maximization**:
>    We solve for the policy $\pi\_r(y|x)$ within the ambiguity set around $\pi\_{\text{ref}}(y|x)$:
>    $$\pi\_r(y|x) = \arg\max\_{\pi\_r} \mathbb{E}\_{x \sim \mathcal{O}, y \sim \pi\_r(y|x)}[r\_\phi(x, y)] \quad s.t. \mathbb{D}\_\phi(\pi\_r(y|x), \pi\_{\text{ref}}(y|x)) \leq \eta.$$
>    This ensures that the reward function $r\_\phi(x, y)$ and the policy $\pi\_r$ are optimized within the divergence constraint.
> 2. **Outer Minimization**:
>    The parameters $\theta$ are then optimized by minimizing the loss function derived from the Bradley-Terry (BT) model:
>    $$\theta = \arg\min\_{\theta} -\mathbb{E}\_{(x, y\_w, y\_l) \sim \mathcal{O}} \left[\log \sigma \left(r\_\phi(x, y\_w) - r\_\phi(x, y\_l)\right)\right].$$
> Here, $r\_\phi(x, y)$ represents the reward function obtained from the inner maximization process. Thus, while our approach does not explicitly solve a separate $\min\max$ structure, it achieves a similar two-level optimization.
>
> Note:
> - In RLHF, the policy $\pi\_\theta$ is updated through RL (PPO), and $r\_\phi$ is updated through Reward Modeling (BT model). In contrast, DPO integrates these two steps. It performs an inner maximization over $\pi_r(y|x)$, allowing $\pi_r$ to represent $r_\phi$. This is followed by an outer minimization with respect to $\theta$, parameterizing $\pi_r$ as $\pi_\theta$.
> - Our aim is not to force a connection between DRO and DPO, but to provide a DRO perspective to explain the role of $\beta$ in DPO.
> ## Why Not the Traditional $\min \max$ Form?
> There are several reasons why the traditional $\min\max$ form is not explicitly used in our method:
> 1. **Efficient Representation Through Analytical Maximization**:
>    - The traditional DRO formulation involves a costly $\max$ operation over an arbitrary distribution $Q'$ within the ambiguity set. However, in our framework, we leverage the problem's structure (e.g., convexity) to analytically solve the inner maximization.
>    - By incorporating the results of the inner maximization into the loss function, we eliminate the need to explicitly perform the $\min\max$ optimization.
>
> 2. **Alignment with Preference Optimization**:
>    - Our method is closely tied to preference learning models (e.g., DPO), where the primary goal is to optimize a preference-based loss rather than explicitly adjust distributions. The $\min\max$ form is reinterpreted in terms of optimizing rewards ($r\_\phi(x, y)$) under divergence constraints, which is more computationally feasible.
>
> 3. **Theoretical Contributions Focus on Inner Maximization**:
>    -  As an example, consider the classic paper [1] on DRO with KL divergence. In its Section 2.1, titled "Solving the Inner Maximization Problem," the analysis focuses on addressing the inner maximization challenge. Similarly, our Theorem 3.1 and Lemma 3.2 are derived based on the Inner Maximization Problem without additional assumptions, allowing independent optimization.
>
> Beyond the analysis above, our contribution includes Dr. DPO: we offer an enhanced version of DPO that additionally achieves pairwise robustness, broadening its applicability and robustness attributes.
>
> [1] Hu Z, Hong L J. Kullback-Leibler divergence constrained distributionally robust optimization[J]. Available at Optimization Online, 2013, 1(2): 9.

---

> > ### Comment · Reviewer_SaBc · 2024-12-01
> > **About pointwise noise**
> >
> > Thank you to the authors for their detailed response. While I still believe that using the DRO framework to explain DPO is far-fetched, I recognize that the primary contribution of this paper lies in the pairwise Dr.DPO framework. As a result, I have raised my score to 6, which will be my final decision. Best of luck to the authors!

---

> > > ### Author Response · Authors · 2024-12-02
> > > **We sincerely appreciate each valuable comment and suggestion.**
> > >
> > > We sincerely thank you for your support and for raising the score of our work. We deeply appreciate your recognition of our contributions to the pairwise Dr. DPO framework. Regarding your valuable suggestions on improving the presentation, we will incorporate them into the final version. Thank you again for your thoughtful feedback, which has been instrumental in enhancing the quality of our work.

---

### Official Review · Reviewer_7Q2Q · 2024-10-31

**Soundness:** 4
**Presentation:** 2
**Contribution:** 3
**Rating:** 6
**Confidence:** 3

**Summary:**

This paper addresses the problem of noise in training datasets for Direct Preference Optimization (DPO). It first classifies noise into pointwise (low-quality data points) and pairwise (erroneous data pair labeling). Then, it analyzes DPO from the perspective of Distributionally Robust Optimization (DRO) to evaluate its resistance to both types of noise. Finally, it presents the Distributionally Robustifying Direct Preference Optimization (Dr. DPO) framework, which incorporates pairwise robustness by optimizing for worst-case pairwise situations and features a new hyperparameter β' to manage data pair reliability. Experiments reveal that Dr. DPO enhances the quality of the generated text within preference datasets.

**Strengths:**

- DPO is inherently a form of Distributionally Robust Optimization (DRO) that confers pointwise robustness, with the regularization coefficient $\beta$ inversely related to the robustness radius $\eta$.
- The authors introduce Distributionally Robustifying DPO (Dr. DPO), which incorporates pairwise robustness by optimizing against the worst-case pairwise scenarios, without the need for explicit noise estimation.
- Dr. DPO demonstrates enhanced performance compared to DPO and other baselines in both noisy and noise-free environments, showcasing its effectiveness in aligning language models with human preferences.

**Weaknesses:**

- The assumptions about pointwise and pairwise noise in Subsection 3.1 and Subsection 4.1 are not quite comparable, which may affect the conclusion in the paper that DPO is robust to pointwise noise but not robust to pairwise noise. Details in Question 2.
- There is a lack of ablation experiments regarding the batch size in the paper. Details in Question 4.
- Some declarations in the paper is not so clear, which make it a little hard to follow. For example, 1) It is declared a topic "DPO is Implicitly a Pointwise DRO" in subsection A on line 192. But apart from a standalone Theorem 3.1 and some "insights" statements, there are lacking of a formal proof to connect DPO and Pointwise RPO. 2) When discussing "Why DPO is Robust to Pointwise Noise" on line 212, it is directly stated that "the implicit robust framework of DPO counters data perturbations effectively" on line 215, rising the question again "why is DPO robust to pointwise perturbations?". As a comparison, Section 4.3 is presented much better.

**Questions:**

1. In Figure 4 (Left), there seem to be similar trends of trading off between Reward and KL, and if a small KL value is required under the scenario of RLHF (e.g. less than 2), are similar rewards gained from different rates of flipped pairs?

2. The noise added to the pointwise dataset and the pairwise dataset in the paper cannot be regarded as the same type of noise. The noise added to the pointwise dataset by generating samples from the un-tuned GPT-2 only weakens the extent of a sample being good, similar to reducing $P(y\ \text{is\ positive})$ rather than setting $P(y\ \text{is\ positive})=0$. So a more corresponding method of adding noise to the pairwise dataset would be reducing $P(y_w > y_l)$, rather than reversing the order to make $P(y_w > y_l)=0$. Therefore, it seems that the different impacts on DPO caused by adding noise to the pointwise and pairwise datasets in Sections 3.1 and 4.1 are not really comparable.

3. The toy example in Section 4.3 is a little hard to follow. The function $h$ that has been clearly defined in the previous text is used, but an inconsistent combination of parameters is passed in, which is confusing.

4. During the training process, the estimations of both $L_{Dr.DPO}$ and $\omega$ rely on a batch of data. Will the batch size become a crucial factor affecting the effectiveness of the policy? There is a lack of ablation experiments regarding the batch size in the paper.

PS. There is a typo "DRDPO" in Section 5.3.

---

> ### Author Response · Authors · 2024-11-22
>
> We greatly appreciate your thoughtful feedback and insightful suggestions. Below, we address your concerns in detail, aiming to clarify and improve our paper.
>
> ---
> **Q1: The assumptions about pointwise and pairwise noise in Subsection 3.1 and Subsection 4.1 are not quite comparable, which may affect the conclusion in the paper that DPO is robust to pointwise noise but not robust to pairwise noise.**
>
> **A1:** We define pointwise and pairwise noise based on practical considerations during fine-tuning large models:
>
> 1. **Pointwise Noise** refers to low-quality data points containing irrelevant or incoherent information. This type of noise can undermine the reliability of the initialization model $\pi_{\text{SFT}}$ during the DPO phase.
> 2. **Pairwise Noise** arises from erroneous associations between data pairs, leading to misjudged preference rankings. This type of noise is likely to be amplified during the DPO phase, thereby causing overfitting issues related to label flipping.
>
>
> Furthermore, **we emphasize that pointwise and pairwise noise are inherent to different types of noise and are not meant to be directly comparable.** Our statement that "DPO is robust to pointwise noise" is grounded in its alignment with the DRO objective (Section 3.2). While pointwise noise may lead to suboptimal initialization of $\pi_{\theta,0} = \pi_{\text{SFT}}$, adjusting $\beta$ enables better optimization toward a reliable $\pi_\theta$. Conversely, pairwise noise directly impacts DPO's objective function (Section 4.3), which Dr. DPO aims to address.
>
>
> **Q2: There is a lack of ablation experiments regarding the batch size in the paper.**
>
> **A2:** To examine the effect of batch size on optimization results, we conducted additional experiments using Llama3-8B-Instruct as the base model. Preference optimization was performed on the UltraFeedback dataset, and evaluations were conducted on the AlpacaEval2 benchmark. The results are presented below:
>
> | **Method (Noise Ratio = 0.0)** | **Length-Control Win Rate** | **Raw Win Rate** |
> |--------------------------------|-----------------------------|-------------------|
> | DPO (batch size = 4)           | 41.81                      | 39.96            |
> | Dr. DPO (batch size = 4)         | 43.47                      | 41.16            |
> |                                |                             |                   |
> | DPO (batch size = 8)           | 40.65                      | 38.63            |
> | Dr. DPO (batch size = 8)         | 42.21                      | 39.79            |
> |                                |                             |                   |
> | DPO (batch size = 16)          | 39.13                      | 36.12            |
> | Dr. DPO (batch size = 16)        | 41.21                      | 38.04            |
>
> | **Method (Noise Ratio = 0.2)** | **Length-Control Win Rate** | **Raw Win Rate** |
> |--------------------------------|-----------------------------|-------------------|
> | DPO (batch size = 4)           | 34.93                      | 33.66            |
> | Dr. DPO (batch size = 4)         | 37.48                      | 36.71            |
> |                                |                             |                   |
> | DPO (batch size = 8)           | 35.38                      | 33.60            |
> | Dr. DPO (batch size = 8)         | 37.09                      | 35.05            |
> |                                |                             |                   |
> | DPO (batch size = 16)          | 36.38                      | 34.41            |
> | Dr. DPO (batch size = 16)        | 38.23                      | 35.66            |
>
> **Notes:**
> - The "batch size" in the tables refers to the number of samples used in a single computation of $\mathbb{E}_O$. Gradient accumulation was applied with `gradient_accumulation_steps = 16`, so the effective batch sizes are 64 (4×16), 128 (8×16), and 256 (16×16).
> - Observations:
>   1. Batch size significantly impacts DPO performance. In noise-free scenarios, larger batch sizes tend to reduce performance, which might be attributed to characteristics of Llama3-8B-Instruct (refer to [Issue](https://github.com/princeton-nlp/SimPO/issues/61#issuecomment-2297040967)).
>   2. Dr. DPO consistently demonstrates improved performance across all batch sizes, particularly in noisy scenarios, where the performance gains are more pronounced.

---

> > ### Comment · Reviewer_7Q2Q · 2024-11-26
> >
> > A larger batch size leads to fewer parameter update steps, requiring adjustments to the learning rate[1]. If these adjustments are not made, performance may decline. Therefore, it is inaccurate to claim directly that "in noise-free scenarios, larger batch sizes tend to reduce performance."
> >
> > Leaving this aside, the precision of the estimate for $\omega$ should improve with a larger batch size, as indicated in Eq. 47. However, our experiments did not provide any supporting evidence for this.
> >
> > [1] Smith, S. L. "Don't decay the learning rate, increase the batch size." arXiv preprint arXiv:1711.00489 (2017).

---

> > > ### Author Response · Authors · 2024-11-26
> > >
> > > **Q6: Therefore, it is inaccurate to claim directly that "in noise-free scenarios, larger batch sizes tend to reduce performance."**
> > >
> > > **A6:** It is important to clarify that the statement "larger batch sizes tend to reduce performance" is not our assertion, but rather an observed phenomenon. This observation aligns with findings from related studies on DPO. For instance, [Issue in SimPO](https://github.com/princeton-nlp/SimPO/issues/61#issuecomment-2297040967) reports similar occurrences, attributing them to:
> > > > 1. Llama 3 instruct models being prone to catastrophic forgetting.
> > > > 2. Larger batch sizes being less effective in learning PO, which helps preserve MMLU and GSK performance from the original Llama 3 instruct model.
> > >
> > > We appreciate your suggestion regarding learning rate adjustments; this is indeed a valuable point for further exploration.
> > >
> > > **Q7: However, our experiments did not provide any supporting evidence for this.**
> > >
> > > **A7:** Thank you for your feedback. In noisy scenarios, we observed a clear trend: larger batch sizes lead to higher precision in estimating $\omega$, thereby enhancing DPO performance. This is consistent with the core motivation proposed by Dr. DPO in contexts involving pairwise noise. For noise-free scenarios, we hypothesize, as mentioned in A6, that the characteristics of the Llama 3 instruct model and the lack of learning rate adjustments may be contributing factors. **We are currently conducting further experiments (using a Mistral-based model and adjusting the learning rate) and will report  progress promptly.**
> > >
> > > We appreciate your suggestions, which we believe will enhance the quality of this work. Additionally, we emphasize that these issues do not affect the core contributions of our study.

---

> > > > ### Comment · Reviewer_7Q2Q · 2024-11-27
> > > >
> > > > Thanks for your response. I keep my overall score for the moment.

---

> > > > > ### Author Response · Authors · 2024-12-02
> > > > >
> > > > > Thank you for your valuable suggestion. We have conducted more detailed experiments to analyze [the effects of batch size](https://github.com/princeton-nlp/SimPO/issues/61#issuecomment-2297040967), and the results are as follows:
> > > > >
> > > > > To minimize the influence of batch size on learning rate and to address the issue of `Llama-3-8B-Instruct` models being prone to catastrophic forgetting, we followed the setting of [1][2][3][4], employed the `Llama-3-8B-Base model`. The learning rate was tuned within the range $[3\mathrm{e}{-7}, 5\mathrm{e}{-7}, 8\mathrm{e}{-7}, 1\mathrm{e}{-6}, 2\mathrm{e}{-6}]$, and we evaluated the performance using the AlpacaEval2 benchmark. The results are summarized below:
> > > > >
> > > > >
> > > > > | **Llama-3-8B-Base model (Noise Ratio = 0.0)** | **Length-Control Win Rate** | **Raw Win Rate** |
> > > > > |--------------------------------|-----------------------------|-------------------|
> > > > > | DPO (batch size = 4)           | 18.13                      | 17.21          |
> > > > > | Dr. DPO (batch size = 4)         | 19.19                      | 18.38            |
> > > > > |                                |                             |                   |
> > > > > | DPO (batch size = 8)           | 18.99                      | 17.46             |
> > > > > | Dr. DPO (batch size = 8)         | 20.32                      | 19.05            |
> > > > > |                                |                             |                   |
> > > > > | DPO (batch size = 16)          | 18.74                      | 17.67            |
> > > > > | Dr. DPO (batch size = 16)        | 21.58                      | 20.31            |
> > > > >
> > > > > | **Llama-3-8B-Base model (Noise Ratio = 0.2)** | **Length-Control Win Rate** | **Raw Win Rate** |
> > > > > |--------------------------------|-----------------------------|-------------------|
> > > > > | DPO (batch size = 4)           | 14.78                      | 12.96            |
> > > > > | Dr. DPO (batch size = 4)         | 16.68                      | 14.30            |
> > > > > |                                |                             |                   |
> > > > > | DPO (batch size = 8)           | 14.59                      | 12.56            |
> > > > > | Dr. DPO (batch size = 8)         | 17.31                      | 14.78            |
> > > > > |                                |                             |                   |
> > > > > | DPO (batch size = 16)          | 13.64                      | 12.12            |
> > > > > | Dr. DPO (batch size = 16)        | 17.55                      | 15.15            |
> > > > >
> > > > > Key observations:
> > > > > 1. **When the learning rate is independently adjusted for each batch size, the DPO method exhibits comparable performance across different batch sizes.** This validates the reviewer's suggestion that `a larger batch size leads to fewer parameter update steps, requiring adjustments to the learning rate`.
> > > > >
> > > > > 2. **As the batch size increases, the performance of Dr. DPO improves consistently.** This trend holds true in both noise-free scenarios and those with a noise ratio of 20%. This confirms that larger batch sizes lead to more accurate estimations of $\omega(x, y_w, y_l)$, resulting in a significant enhancement in model performance.
> > > > >
> > > > > 3. **Performance improvements are also evident in the noise-free scenario, further emphasizing the necessity of Dr. DPO.** In the 0% flipped case (original dataset without intentional label flips), Dr. DPO continues to outperform DPO. This result suggests the presence of inherent label noise in existing datasets, which serves as a core motivation for Dr. DPO. This finding aligns with the observations in rDPO [5], where a default flip rate of 0.1 was shown to improve performance on the HH dataset, as reported in Table 3 of their work.
> > > > >
> > > > > We sincerely appreciate your suggestions regarding our work, and we look forward to integrating these improvements into our manuscript. We are grateful for your contributions to refining Dr. DPO during this rebuttal phase!
> > > > >
> > > > > **References:**
> > > > > [1] Sen Yang, Leyang Cui, Deng Cai, Xinting Huang, Shuming Shi, Wai Lam. Not All Preference Pairs Are Created Equal: A Recipe for Annotation-Efficient Iterative Preference Learning. EMNLP 2024.
> > > > >
> > > > > [2] Yu Meng, Mengzhou Xia, Danqi Chen. SimPO: Simple Preference Optimization with a Reference-Free Reward. NeurIPS 2024.
> > > > >
> > > > > [3] Jiaqi Han, Mingjian Jiang, Yuxuan Song, Jure Leskovec, Stefano Ermon, Minkai Xu. f-PO: Generalizing Preference Optimization with f-divergence Minimization. CoRR abs/2410.21662 (2024)
> > > > >
> > > > > [4] Xiao Yu, Qingyang Wu, Yu Li, Zhou Yu. LIONs: An Empirically Optimized Approach to Align Language Models. EMNLP 2024.
> > > > >
> > > > > [5] Chowdhury, S. R., Kini, A., & Natarajan, N. (2024). Provably robust DPO: Aligning language models with noisy feedback. *ICML 2024*.

---

> ### Author Response · Authors · 2024-11-22
>
> **Q3: Some declarations in the paper are not clear, making it a little hard to follow.**
>
> **A3:** Thank you for highlighting this concern. We provide clarification for the following points:
>
> 1. **"DPO is Implicitly a Pointwise DRO"**
>    - This statement is derived from the theoretical equivalence between DRO and DPO established in Theorem 3.1. In Equation 7, the RM-DRO objective addresses the robust optimization goals of reward modeling, and Theorem 3.1 connects this objective to DPO. Therefore, we state that "DPO is implicitly a Pointwise DRO."
>    - To further elucidate, we have added a detailed connection between DRO and DPO in `Appendix B.2` and `Appendix C.2`, highlighting their shared optimization framework and differing motivations.
>
>
> 2. **"Why DPO is Robust to Pointwise Noise"**
>    - By definition, DRO mitigates pointwise noise through robust optimization (Section 3.2). Since Theorem 3.1 demonstrates the equivalence between DPO and DRO, we conclude that "DPO is robust to pointwise noise."
>
>    - The part of `B. The Optimal Value of $\beta$ Reflects the Noise Level within the SFT Model.` in Section 3.2, offers an alternative perspective on why DPO is robust to pointwise noise. This interpretation is supported by its theoretical relationship with the robustness radius and corroborated by experimental evidence.
>
>
> We hope these additions address your concerns.
>
> **Q4: In Figure 4 (Left), there seem to be similar trends of trading off between Reward and KL. If a small KL value is required under the scenario of RLHF (e.g., less than 2), are similar rewards gained from different rates of flipped pairs?**
>
> **A4:** Figure 4 (Left) illustrates the trade-off between reward and KL under different label-flipping scenarios. The trends suggest that for similar KL values, corresponding rewards are also similar.
>
> In the context of RLHF, if we focus on the single-step training process where the policy model samples responses and the reward model assigns scores, potentially causing label flipping, we posit that its optimization behavior aligns with that of DPO, and the overall trend appears consistent. However, since RLHF involves a multi-step reinforcement learning process, we believe this observed trend may not generalize across iterations. We plan to explore this issue further in future research.
>
> **Q5: The toy example in Section 4.3 is a little hard to follow. The function \(h\) has been clearly defined in the previous text, but an inconsistent combination of parameters is passed in, which is confusing.**
>
> **A5:** We sincerely apologize for this oversight. The inconsistency is due to a typographical error, which has been corrected in the revised manuscript. The updated and more precise formulation is as follows:
>
> $$
> [h(x_1, y_{1,w}, y_{1,l}), h(x_2, y_{2,w}, y_{2,l})] = [-0.1, -1.0].
> $$
>
> This correction ensures consistency with the earlier definitions and clarifies the example.
>
> ---
> We hope our responses address your concerns comprehensively. Thank you once again for your constructive feedback and the opportunity to improve our work.

---

### Official Review · Reviewer_Te5v · 2024-11-03

**Soundness:** 3
**Presentation:** 3
**Contribution:** 3
**Rating:** 5
**Confidence:** 4

**Summary:**

In this paper, the authors explore the noisy data challenges when using Direct preference optimization (DPO) methods in large language models (LLMs). In this paper, noise is divided into pointwise noise and pairwise noise, and a method called Dr. DPO based on DRO is proposed to enhance DPO's resistance to these two kinds of noise. By introducing a new hyperparameter $\beta'$, Dr. DPO is able to achieve a better balance between exploration and utilization in noisy environments. The experimental results show that Dr. DPO not only performs well on noisy data sets, but also shows stronger performance on noiseless data sets.

**Strengths:**

1. The Dr. DPO framework proposed in this paper is an important improvement over the existing DPO methods, which effectively solves the noise problem in training data and improves the robustness and accuracy of the model.

2. The authors provide a solid theoretical basis to explain why DPO essentially incorporates the principles of DRO and how to improve the model's resistance to point noise by adjusting the regularization coefficient $\beta$.

3. This paper verifies the effectiveness of Dr. DPO through a large number of experiments, and the experimental results show that the method can significantly improve the quality of generated text and the accuracy of response under different types of noise environments.

4.  The authors provide links to the code, which helps other researchers reproduce the experimental results and further promote the research progress in related fields.
defect.

5. The authors provide a Toy Example of How Dr. DPO Works, which is helpful to understand the proposed method.

**Weaknesses:**

1. The evaluation could be compared with more baselines, please refers to https://arxiv.org/pdf/2409.02795.

2. These evaluation tasks are too simple; some methods may be effective on benchmarks like IMDB but may not generalize well to more complex tasks.

3. Although the authors provide beautiful theory proof, the objective of Eq (12) seems to be in contradiction with IPO.

**Questions:**

1. Despite how funny this question might sound, can I understand Dr. DPO as simply adding an exponential function ($e^{(\cdot)}$)to the objective function of DPO?

2. Could you explain the weakness-3?  IPO learns the $h(x,y_w,y_l)$ to a specific value, but Dr. DPO maximize the $h(x,y_w, y_l)$, are you be in contradiction ?

---

> ### Author Response · Authors · 2024-11-22
>
> We sincerely thank you for your valuable feedback and thoughtful suggestions. Below, we provide detailed responses to the concerns and questions raised.
>
> ---
>
> **Q1: The evaluation could be compared with more baselines, please refer to [arXiv:2409.02795](https://arxiv.org/pdf/2409.02795).**
>
> **A1:** We appreciate the suggestion to expand the scope of baselines. However, it is important to emphasize that the primary goal of Dr. DPO is not to develop a method that simply excels on benchmarks but to address the issue of pairwise noise. For this reason, we chose baselines that specifically target similar challenges, such as cDPO [1] and rDPO [2].
>
> Additionally, we extended the concept of Dr. DPO to SimPO[3] (a SOTA variant of DPO). Below, we provide results comparing SimPO and Dr. SimPO under noise-free and noisy conditions (Noise Ratio = 0.2), using Llama3-8B-Instruct as the base model and the UltraFeedback dataset for preference optimization. And we evaluate their performance on the AlpacaEval2 benchmark. The results are as follows:
>
> | **Method**           | **Length-Control Win Rate** | **Raw Win Rate** |
> |-----------------------|-----------------------------|------------------|
> | SimPO (Noise Ratio = 0.0)   | 42.89                      | 37.30           |
> | Dr. SimPO (Noise Ratio = 0.0)| 43.26                      | 37.79           |
> |                       |                             |                  |
> | SimPO (Noise Ratio = 0.2)   | 36.69                      | 33.61           |
> | Dr. SimPO (Noise Ratio = 0.2)| 41.27                      | 39.04           |
>
>
> These results demonstrate that the DRO concept underlying Dr. DPO can be effectively extended to DPO-like methods, with particularly significant performance improvements in noisy scenarios. We hope this additional comparison alleviates your concern regarding the choice of baselines.
>
> **References:**
> [1] Rafailov, R., Sharma, A., Mitchell, E., Manning, C. D., Ermon, S., & Finn, C. (2023). A note on DPO with noisy preferences and relationship to IPO. https://ericmitchell.ai/cdpo.pdf.
>
> [2] Chowdhury, S. R., Kini, A., & Natarajan, N. (2024). Provably robust DPO: Aligning language models with noisy feedback. In ICML 2024.
>
> [3] Yu Meng, Mengzhou Xia, Danqi Chen (2024): SimPO: Simple Preference Optimization with a Reference-Free Reward. In NeurIPS 2024.
>
> **Q2: These evaluation tasks are too simple; some methods may be effective on benchmarks like IMDB but may not generalize well to more complex tasks.**
>
> **A2:**  Thank you for your suggestion. To evaluate the impact of preference optimization methods on downstream tasks, we conducted experiments on a range of more challenging benchmarks listed on the HuggingFace Open Leaderboard. The results, using Llama3-8B-Instruct as the base model, and the UltraFeedback dataset for preference optimization. The results are as follows:
>
> | **Method (Noise Ratio = 0.0)** | **ARC-Challenge** | **TruthfulQA** | **GSM8k** | **Average** |
> |--------------------------------|-------------------|--------------------|-----------|-------------|
> | DPO (Llama3-8B-Instruct)       | 64.08            | 60.77             | 66.94     | 63.93       |
> | Dr. DPO (Llama3-8B-Instruct)     | 65.10            | 63.10             | 67.17      | 65.12       |
>
> | **Method (Noise Ratio = 0.2)** | **ARC-Challenge** | **TruthfulQA** | **GSM8k** | **Average** |
> |--------------------------------|-------------------|--------------------|-----------|-------------|
> | DPO (Llama3-8B-Instruct)       | 63.12            | 59.61             | 69.83      | 64.19       |
> | Dr. DPO (Llama3-8B-Instruct)     | 64.68            | 60.35             | 71.72     | 65.58       |
>
> Additionally, we tested Dr. DPO on the AlpacaEval2 benchmark for further validation:
>
> | **Method**             | **Length-Control Win Rate** | **Raw Win Rate** |
> |-------------------------|-----------------------------|------------------|
> | DPO (Noise Ratio = 0.0) | 40.65                      | 38.63           |
> | Dr. DPO (Noise Ratio = 0.0)| 42.21                      | 39.79           |
> |                         |                             |                  |
> | DPO (Noise Ratio = 0.2) | 35.38                      | 33.60           |
> | Dr. DPO (Noise Ratio = 0.2)| 37.09                      | 35.05           |
>
> Across diverse tasks and benchmarks, Dr. DPO consistently improves performance without introducing significant complexity. These results validate the generalizability and robustness of our method.

---

> ### Author Response · Authors · 2024-11-22
>
> **Q3: Although the authors provide a beautiful theoretical proof, the objective of Eq. (12) seems to be in contradiction with IPO.**
>
> **A3:** Thank you for raising this concern. It is important to clarify that IPO and Dr. DPO address different challenges:
> - IPO aims to mitigate overfitting to data by learning $h(x, y\_w, y\_l)$ to a specific value, focusing on regularization.
> - Dr. DPO addresses the sensitivity of DPO to label-flipped noise by reweighting pairwise data within a batch, reducing the amplification of noisy labels.
>
> Thus, the motivations and application scenarios of these methods differ fundamentally. To further support this distinction, we compared the performance of IPO and Dr. DPO using the UltraFeedback dataset with Llama3-8B-Instruct as the base model:
>
>
> | **Method (Noise Ratio = 0.0)** | **Length-Control Win Rate** | **Raw Win Rate** |
> |--------------------------------|-----------------------------|------------------|
> | DPO (Llama3-8B-Instruct)       | 40.65                      | 38.63           |
> | IPO (Llama3-8B-Instruct)       | 35.90                      | 34.48           |
> | Dr. DPO (Llama3-8B-Instruct)     | 42.21                      | 39.79           |
>
> Note: In the aforementioned experimental setting, the performance of DPO and IPO closely aligns with the results reported in SimPO.
>
> These results demonstrate that IPO does not achieve optimal performance under noise-free settings, whereas Dr. DPO consistently improves over DPO by addressing label-flipping issues. We hope this comparison clarifies the complementary nature of these methods.
>
>
> **Q4: Despite how funny this question might sound, can I understand Dr. DPO as simply adding an exponential function ($e^{\cdot}$) to the objective function of DPO?**
>
> **A4:**  A more precise formulation is that Dr. DPO modifies the DPO loss function from $-\mathbb{E}[h(x, y\_w, y\_l)]$ to $-\beta' \log \mathbb{E}\left[\exp\left(\frac{h(x, y\_w, y\_l)}{\beta'}\right)\right]$, introducing a `Log-Expectation-Exp` structure. This structure enables the method to address label-flipping issues effectively.
>
> The gradient formulation highlights the difference:
> $$ \frac{\partial}{\partial h} \left(-\beta' \log \mathbb{E}\_{\mathcal{O}}\left[\exp\left(\frac{h}{\beta'}\right)\right]\right) = -\frac{\exp\left(\frac{h}{\beta'}\right)}{\mathbb{E}\_{\mathcal{O}}\left[\exp\left(\frac{h}{\beta'}\right)\right]}. $$
>
> This reweighting mechanism assigns higher weights to samples with larger $h(x, y\_w, y\_l)$ when $\beta'$ is small, while converging to a uniform distribution as $\beta'$ increases. Therefore, Dr. DPO generalizes DPO by incorporating this adaptive weighting mechanism.
>
> ---
> We hope these responses address your concerns comprehensively. Thank you again for your thoughtful comments and the opportunity to improve our work.

---

> > ### Comment · Reviewer_Te5v · 2024-11-27
> > **Thanks**
> >
> > Thanks for your response.
> >
> >
> > I still argue that the novelty of this approach is relatively weak, and the theoretical proof lacks practical guidance. For example, the upper bound in eq.(14) (the right-hand side of the inequality) contains $\frac{1}{\delta}$, which is unfavorable for convergence. The final optimization objective is a variant of DPO, and I have concerns about the practical applicability of this method. However, considering the additional experiments and responses provided by the authors, I have decided to stick to my score.

---

> > > ### Author Response · Authors · 2024-11-27
> > >
> > > **Q4: For example, the upper bound in eq.(14) (the right-hand side of the inequality) contains $\frac{1}{\delta}$, which is unfavorable for convergence.**
> > >
> > > **A4:**
> > > We appreciate the reviewer’s feedback and would like to clarify the role of $\frac{1}{\delta}$ in the theoretical bound. **The term $\frac{1}{\delta}$ appears naturally in the context of generalization theory, as it is a standard component of probabilistic confidence bounds.** Specifically, $\delta$ represents the confidence parameter, and $1 - \delta$ corresponds to the probability that the generalization bound holds. This structure is intrinsic to statistical learning theory and is not specific to our method.
> > >
> > > For example, similar forms of bounds containing $\frac{1}{\delta}$ are widely used in Rademacher complexity-based generalization error bounds [1], PAC learnability theory [2], debiasing in recommendation systems [3], and domain adaptation frameworks [4]. In these works, $\frac{1}{\delta}$ quantifies the statistical confidence, ensuring that the theoretical guarantee holds with high probability. Importantly, this term does not directly influence the optimization process or the convergence of the algorithm but instead provides statistical rigor to the analysis. Thus, its inclusion in our theoretical analysis follows well-established practices in the field and does not impact the practical convergence of our method.
> > >
> > > Additionally, the purpose of the upper bound we propose is to demonstrate that as the sample size increases, Dr. DPO optimizes the "ideal" distribution that accurately discerns the correct ranking between pairwise instances. This serves as the theoretical foundation for the success of Dr. DPO and supports its practical utility.
> > >
> > > **Q5: The final optimization objective is a variant of DPO, and I have concerns about the practical applicability of this method.**
> > >
> > > **A5:**
> > > In our work, we have thoroughly evaluated the proposed method across various datasets and benchmarks. Specifically, we experimented with the IMDB and Anthropic HH datasets, aligning with the experimental settings of the original DPO work. In addition, we introduced a new dataset, the UltraFeedback dataset, and assessed our method using multiple benchmarks, including the HuggingFace Open Leaderboard, AlpacaEval2, and MT-bench. Furthermore, we conducted experiments with Dr. DPO across different model types (Llama and Mistral) and model sizes (7B, 8B, 13B). **These experiments, spanning multiple datasets, evaluation benchmarks, model architectures, and noise conditions, demonstrate the robustness and broad applicability of our method.**
> > >
> > > While the final optimization objective is inspired by DRO, our approach introduces modifications that make it both simpler to implement (requiring only a single line of code modification) and more effective in noisy and noise-free scenarios, as validated by extensive experiments. Therefore, we believe that the novelty of this work is sufficient, as it combines ease of implementation with significant theoretical and practical contributions, encapsulating an approach that is both easy and effective.
> > >
> > > **References:**
> > >
> > > [1] Bounding the Generalization Error in Rademacher Complexity. https://en.wikipedia.org/wiki/Rademacher_complexity.
> > > [2] Shai Shalev-Shwartz and Shai Ben-David. 2014. *Understanding Machine Learning: From Theory to Algorithms.* Cambridge University Press.
> > > [3] Jiawei Chen, Hande Dong, Yang Qiu, Xiangnan He, Xin Xin, Liang Chen, Guli Lin, Keping Yang: *AutoDebias: Learning to Debias for Recommendation.* SIGIR 2021: 21-30.
> > > [4] Chao Zhang, Lei Zhang, Jieping Ye: *Generalization Bounds for Domain Adaptation.* NIPS 2012.

---

### Official Review · Reviewer_eoeA · 2024-11-03

**Soundness:** 3
**Presentation:** 3
**Contribution:** 3
**Rating:** 6
**Confidence:** 4

**Summary:**

This paper addresses the challenge of noisy training data in Direct Preference Optimization (DPO) for aligning Large Language Models (LLMs) with human preferences. The authors identify two types of noise: pointwise noise (low-quality data points) and pairwise noise (incorrect preference rankings). They enhance DPO's robustness using Distributionally Robust Optimization (DRO). They reveal that DPO inherently provides robustness to pointwise noise. To tackle pairwise noise, they introduce Distributionally Robustifying DPO (Dr. DPO), which optimizes against worst-case pairwise scenarios and introduces a new hyperparameter to control data pair reliability. Empirical results show that Dr. DPO significantly improves text quality and response accuracy in both noisy and clean datasets, offering a more robust method for LLM alignment in the presence of data noise.

**Strengths:**

1. The paper is well written and easy to follow.
2. Addressing the issue of noisy data in preference learning for LLMs is both important and interesting.
3. The paper has extensive experiments and theoretical analysis to prove the effectiveness of their proposed method.

**Weaknesses:**

1. The authors include IMDB and Anthrophic HH for experiments. It’s better to provide more datasets like TruthfulQA, GSM8k and other datasets.
2. It’s better for authors to conduct experiments on more open-source LLMs to further verify the effectiveness of their algorithms.
3. The authors can provide some more case studies to better understand the performance of their method.

**Questions:**

I have included my questions in the weaknesses part.

---

> ### Author Response · Authors · 2024-11-22
>
> We thank the reviewer for their thoughtful feedback and constructive suggestions. Below, we address your comments in detail.
>
> ---
> **Q1: It is better to provide more datasets like TruthfulQA, GSM8k, and other datasets.**
>
> **A1:**  We appreciate your insightful suggestion. To evaluate the impact of preference optimization methods on downstream task performance, we conducted additional experiments using Llama3-8B-Instruct as the base model. Preference optimization was performed on the UltraFeedback dataset, and evaluations were conducted on the different tasks such as ARC-Challenge, TruthfulQA-MC2,  and GSM8k. The results are summarized below:
>
> | **Method (Noise Ratio = 0.0)** | **ARC-Challenge** | **TruthfulQA** | **GSM8k** | **Average** |
> |--------------------------------|-------------------|--------------------|-----------|-------------|
> | DPO (Llama3-8B-Instruct)       | 64.08            | 60.77             | 66.94     | 63.93       |
> | Dr. DPO (Llama3-8B-Instruct)     | 65.10            | 63.10             | 67.17      | 65.12       |
>
> | **Method (Noise Ratio = 0.2)** | **ARC-Challenge** | **TruthfulQA** | **GSM8k** | **Average** |
> |--------------------------------|-------------------|--------------------|-----------|-------------|
> | DPO (Llama3-8B-Instruct)       | 63.12            | 59.61             | 69.83      | 64.19       |
> | Dr. DPO (Llama3-8B-Instruct)     | 64.68            | 60.35             | 71.72     | 65.58       |
>
> These results demonstrate that Dr. DPO achieves consistent performance improvements across various tasks with just a
> single additional line of code.
>
> Due to time constraints, we primarily explored the noise-free and noise ratio = 0.2 settings. However, the results validate that Dr. DPO is robust to noise and offers stable performance gains across diverse datasets.
>
>
> **Q2: It is better for the authors to conduct experiments on more open-source LLMs to further verify the effectiveness of their algorithms.**
>
> **A2:**  Thank you for this suggestion. We extended our evaluation to include both Llama3-8B-Instruct and Mistral-Instruct-7B as the base models. Preference optimization was performed on the UltraFeedback dataset, and evaluations were conducted on the AlpacaEval2 benchmark. The results are as follows:
>
> | **Method (Noise Ratio = 0.0)** | **Length-Control Win Rate** | **Raw Win Rate** |
> |--------------------------------|-----------------------------|------------------|
> | DPO (Llama3-8B-Instruct)       | 40.65                      | 38.63           |
> | Dr. DPO (Llama3-8B-Instruct)     | 42.21                      | 39.79           |
> |                                |                             |                  |
> | DPO (Mistral-Instruct-7B)      | 20.91                      | 18.90           |
> | Dr. DPO (Mistral-Instruct-7B)    | 22.80                      | 19.77           |
>
> These results confirm that DrDPO consistently improves performance across different model families (Llama and Mistral) and model sizes (7B and 8B), further validating the robustness and generalizability of the proposed approach.
>
>
> **Q3: The authors can provide some more case studies to better understand the performance of their method.**
>
> **A3:**  Thank you for this suggestion. To better illustrate the qualitative impact of our method, we have added several case studies to the `Appendix E` of the revised manuscript. These examples highlight scenarios where DrDPO demonstrates improved text quality and response accuracy compared to baseline methods. We hope these additions provide a clearer understanding of the practical benefits of our approach.
>
> ---
> We hope these additional results and clarifications address your concerns comprehensively. Thank you again for your thoughtful review and the opportunity to improve our work.

---

> > ### Comment · Reviewer_eoeA · 2024-11-24
> >
> > Thanks for your rebuttal, which solves most of my concerns. I will keep my overall score.

---

### Official Review · Reviewer_dGhR · 2024-11-04

**Soundness:** 4
**Presentation:** 4
**Contribution:** 4
**Rating:** 8
**Confidence:** 3

**Summary:**

The paper addresses the noisy data problem in DPO alignment. Specifically, the authors start with pointwise noise data and show that the SFTed model on these data harms DPO performance. Then, the authors build a connection between DPO and DRO and point out that the smaller $\beta$ in DPO implicitly enlarges the search space of DRO, therefore suitable for larger noise.

Authors then argue that DPO cannot handle the pairwise noise and show pairwise noise harm the DPO performance. To tackle the problem, authors derive DrDPO from DRO. DRO can be understood as a dynamic-reweighting DPO that assigns more weight to less likely noisy data.

During experiments, authors show how small $\beta$ can mitigate pointwise noisy data and DrDPO achieve higher rewards than DPO with faster convergence and similar KL.

**Strengths:**

* Clear problem formulation and motivation. Each motivation has introductory supporting experiments.

* Concise writing is easy for readers to understand even though readers do not know DRO much in advance. The running example is very helpful to understand the algorithm.

* Extensive experiments and ablation study to support motivation, main claim (DrDPO), and method's characteristics (convergence, KL, etc).

**Weaknesses:**

* DrDPO requires the estimation of E_O.  In practice, this is often achieved within a batch or pre-computed on the whole dataset. Which implementation is used in the paper? How does the batch size affect the optimization results if using the former one?

* Experiments on more models and sizes will make the results more convincing.

* There are several papers discussing reweighting DPO, such as WPO (https://arxiv.org/pdf/2406.11827v1), which could be discussed in related work.

**Questions:**

Typo: line 194: \pi_\theta

---

> ### Author Response · Authors · 2024-11-22
>
> We sincerely thank you for acknowledging our contributions and for providing valuable feedback. Below, we address each of your comments in detail.
>
> ---
> **Q1: Which implementation is used in the paper? How does the batch size affect the optimization results if using the former one?**
>
> **A1:**
> In the paper, Dr. DPO computes the estimation of $\mathbb{E}\_O$ within a batch. To examine the effect of batch size on optimization results, we conducted additional experiments using Llama3-8B-Instruct as the base model. Preference optimization was performed on the UltraFeedback dataset, and evaluations were conducted on the AlpacaEval2 benchmark. The results are presented below:
>
> | **Method (Noise Ratio = 0.0)** | **Length-Control Win Rate** | **Raw Win Rate** |
> |--------------------------------|-----------------------------|-------------------|
> | DPO (batch size = 4)           | 41.81                      | 39.96            |
> | Dr. DPO (batch size = 4)         | 43.47                      | 41.16            |
> |                                |                             |                   |
> | DPO (batch size = 8)           | 40.65                      | 38.63            |
> | Dr. DPO (batch size = 8)         | 42.21                      | 39.79            |
> |                                |                             |                   |
> | DPO (batch size = 16)          | 39.13                      | 36.12            |
> | Dr. DPO (batch size = 16)        | 41.21                      | 38.04            |
>
> | **Method (Noise Ratio = 0.2)** | **Length-Control Win Rate** | **Raw Win Rate** |
> |--------------------------------|-----------------------------|-------------------|
> | DPO (batch size = 4)           | 34.93                      | 33.66            |
> | Dr. DPO (batch size = 4)         | 37.48                      | 36.71            |
> |                                |                             |                   |
> | DPO (batch size = 8)           | 35.38                      | 33.60            |
> | Dr. DPO (batch size = 8)         | 37.09                      | 35.05            |
> |                                |                             |                   |
> | DPO (batch size = 16)          | 36.38                      | 34.41            |
> | Dr. DPO (batch size = 16)        | 38.23                      | 35.66            |
>
> **Notes:**
> - The "batch size" in the tables refers to the number of samples used in a single computation of $\mathbb{E}\_O$. Gradient accumulation was applied with `gradient_accumulation_steps = 16`, so the effective batch sizes are 64 (4×16), 128 (8×16), and 256 (16×16).
> - Observations:
>   1. Batch size significantly impacts DPO performance. In noise-free scenarios, larger batch sizes tend to reduce performance, which might be attributed to characteristics of Llama3-8B-Instruct (refer to [Issue](https://github.com/princeton-nlp/SimPO/issues/61#issuecomment-2297040967)).
>   2. Dr. DPO consistently demonstrates improved performance across all batch sizes, particularly in noisy scenarios, where the performance gains are more pronounced.
>
>
> **Q2: Experiments on more models and sizes will make the results more convincing.**
>
> **A2:** In Table 3 of the original manuscript, we presented results using Llama2-13B. In response to your comment, we have included experiments with the Llama3-8B-Instruct model under different batch sizes (detailed in Q1). Additionally, we have extended our evaluation to Mistral-Instruct-7B, using AlpacaEval2 as the benchmark. The results are as follows:
>
> | **Method (Noise Ratio = 0.0)** | **Length-Control Win Rate** | **Raw Win Rate** |
> |--------------------------------|-----------------------------|-------------------|
> | DPO (Llama3-8B-Instruct)       | 40.65                      | 38.63            |
> | Dr. DPO (Llama3-8B-Instruct)     | 42.21                      | 39.79            |
> |                                |                             |                   |
> | DPO (Mistral-Instruct-7B)      | 20.91                      | 18.90            |
> | Dr. DPO (Mistral-Instruct-7B)    | 22.80                      | 19.77            |
>
> These results highlight the consistent performance gains of Dr. DPO across both model types (Llama and Mistral) and varying model sizes (7B, 8B, 13B).
>
>
> **Q3: There are several papers discussing reweighting DPO, such as WPO ([arXiv:2406.11827v1](https://arxiv.org/pdf/2406.11827v1)), which could be discussed in related work.**
>
> **A3:**
> Thank you for this suggestion. In the revised manuscript, we have incorporated a discussion of reweighting DPO methods, including WPO and related work. We appreciate your recommendation, which has helped us strengthen the related work section.

---

> ### Author Response · Authors · 2024-11-22
>
> **Q4: Typo: line 194: \pi_\theta**
>
> **A4:**
> We greatly appreciate your meticulous review. The typographical error in line 194 (\pi_\theta) has been corrected in the revised manuscript.
>
> ---
> We hope that our responses and the additional experiments adequately address your concerns. Thank you once again for your constructive feedback and the opportunity to improve our work.

---

> > ### Comment · Reviewer_dGhR · 2024-11-23
> >
> > Thanks, authors, for the rebuttal. The authors do address my concerns, and I will keep my score.

---

> > > ### Author Response · Authors · 2024-11-24
> > >
> > > Thank you for maintaining your score after our rebuttal. Your positive feedback means a great deal to us and validates the effort we put into our work and our rebuttal.

---

### Author Response · Authors · 2024-12-04

We thank all reviewers for their valuable and insightful feedback.

We are encouraged that the reviewers found our paper meaningful (Reviewers $\color{red}{\text{dGhR}}$, $\color{blue}{\text{eoeA}}$, $\color{green}{\text{Te5v}}$, $\color{black}{\text{7Q2Q}}$, $\color{orange}{\text{SaBc}}$). Furthermore, we are grateful that the reviewers recognized the simplicity and effectiveness of our proposed Dr. DPO algorithm (Reviewers $\color{red}{\text{dGhR}}$, $\color{blue}{\text{eoeA}}$, $\color{green}{\text{Te5v}}$, $\color{black}{\text{7Q2Q}}$, $\color{orange}{\text{SaBc}}$). We also appreciate that several reviewers found our paper well-written and easy to follow (Reviewers $\color{red}{\text{dGhR}}$, $\color{blue}{\text{eoeA}}$, $\color{orange}{\text{SaBc}}$).

We acknowledge the reviewers' constructive comments and critiques, which have helped us identify areas for improvement. Below, we summarize our discussions and detailed responses to each reviewer’s feedback:

- **Reviewer $\color{red}{\text{dGhR}}$**: We have added experiments on batch size to validate its relationship with Dr. DPO and included results with more model sizes and types. Considering your positive rating, we believe your concerns have been addressed. Thank you for your encouraging feedback.

- **Reviewer $\color{blue}{\text{eoeA}}$**: We included additional benchmarks, such as TruthfulQA and GSM8k, as well as results for different model types to further validate the effectiveness of Dr. DPO. Considering your positive rating, we believe your concerns have been addressed. Thank you for your encouraging feedback.

- **Reviewer $\color{green}{\text{Te5v}}$**: We incorporated additional benchmarks, including TruthfulQA, GSM8k, and AlpacaEval2, along with evaluations of a variant (Dr. SimPO), to validate the effectiveness of Dr. DPO. Furthermore, we provided relevant literature support for the $\frac{1}{\delta}$ term appearing in the bound. Thank you for your constructive comments.

- **Reviewer $\color{black}{\text{7Q2Q}}$**: We conducted more detailed experiments, including evaluations with llama3-8-base and varying batch sizes under different learning rates, to validate their relationship with Dr. DPO. Considering your positive rating, we believe your concerns have been addressed. Thank you for your encouraging feedback.

- **Reviewer $\color{orange}{\text{SaBc}}$**: We provided an analysis of pointwise noise and its connection to DRO. Additionally, we emphasized the pairwise Dr. DPO framework as a core contribution of the paper, demonstrating its theoretical and experimental effectiveness. Considering your positive rating, we believe your concerns have been addressed. Thank you for your constructive feedback.

We are pleased to note that Reviewers $\color{red}{\text{dGhR}}$, $\color{blue}{\text{eoeA}}$, $\color{black}{\text{7Q2Q}}$, and $\color{orange}{\text{SaBc}}$ provided positive ratings and that their concerns have been addressed. For Reviewer $\color{green}{\text{Te5v}}$, we understand that you may have been too busy to participate in the discussion phase. However, we kindly request that you re-evaluate our paper in the subsequent stages.

As the author-reviewer discussion phase concludes, we sincerely hope that our responses and improvements have effectively addressed your concerns. If there are any remaining questions, we are more than happy to provide further clarifications. Once again, we thank all reviewers for their thoughtful efforts in improving the quality of our work.

---

### Meta-Review · Area_Chair_fMS8 · 2024-12-22

**Metareview:**

This paper tackles noise in alignment data, using the natural idea of applying distributionally robust optimization to alignment techniques. The authors study certain types of noise models, noting that some of them are inherent in the alignment (e.g., DPO) objective, while others are more complex and so can benefit from DRO. They show a simple model for noise and theoretical result when using DRO. Then they perform various experiments, focusing particularly on cases where noise has been injected (i.e., a certain percentage of preference pairs are flipped). In these cases, vanilla DPO struggles while DRO+DPO performs better, as expected. The authors also study general datasets that are likely to inherently have noise, showing good results over baselines.

The strengths for this work are a simple but well-executed idea, strong writing, and solid results. In terms of weaknesses, alignment improvements are notoriously hard to generalize across models, datasets, techniques, etc. so seeing more results/baselines/experimental settings would have been helpful. To their credit the authors did include more in the rebuttal phase.

Overall this was a reasonably strong paper that performs a natural combination of robust optimization with alignment. It is over the bar for acceptance.

**Additional Comments On Reviewer Discussion:**

Most reviewers agreed the paper is strong, but asked for (i) additional clarity in the writing, particularly in the theoretical areas, and (ii) further experimental evidence. I agreed with the reviewers; the rebuttal was helpful in that the authors both clarified many of the requested areas and also added more evidence.

---

### Decision · Program_Chairs · 2025-01-22

Accept (Poster)